# Aging and diet alter the protein ubiquitylation landscape in the mouse brain

Antonio Marino [1,2,6], Domenico Di Fraia[1,3,6], Diana Panfilova[1,4], Amit Kumar Sahu [1], Alberto Minetti[1], Omid Omrani [1], Emilio Cirri[1] & Alessandro Ori [1,5] ✉

Post-translational modifications (PTMs) regulate protein homeostasis, but how aging impacts PTMs remains unclear. Here, we used mass spectrometry to reveal changes in hundreds of protein ubiquitylation, acetylation, and phosphorylation sites in the mouse aging brain. We show that aging has a major impact on protein ubiquitylation. 29% of the quantified ubiquitylation sites were affected independently of protein abundance, indicating altered PTM stoichiometry. Using iPSC-derived neurons, we estimated that 35% of ubiquitylation changes observed in the aged brain can be attributed to reduced proteasome activity. Finally, we tested whether protein ubiquitylation in the brain can be influenced by dietary intervention. We found that one cycle of dietary restriction and re-feeding modifies the brain ubiquitylome, rescuing some but exacerbating other ubiquitylation changes observed in old brains. Our findings reveal an age-dependent ubiquitylation signature modifiable by dietary intervention, providing insights into mechanisms of protein homeostasis impairment and highlighting potential biomarkers of brain aging.

Post-translational modifications (PTMs) expand the chemical diversity of proteins by generating distinct proteoforms encoded by the same gene[1]. PTMs modulate proteins' localization, interactions, stability, and turnover, thereby influencing protein homeostasis (proteostasis)[2]. Loss of proteostasis is a hallmark of aging[3–5] and age-related diseases, particularly neurodegenerative disorders[6,7]. Acetylation, phosphorylation, and ubiquitylation are among the most studied PTMs, accounting for more than 90% of all reported modifications to date[8]. Acetylation is fundamental for brain gene expression, regulating the accessibility of histones[9]. Histone acetylation is essential for neuronal maturation, synapse formation, and establishment of neuronal circuits[10]. Phosphorylation mediates signaling and modulates multiple neuronal functions. For instance, calcium/calmodulin-dependent protein kinase II modulates synaptic strength, mainly by affecting trafficking, function, and anchoring to the postsynaptic membrane of glutamate transmembrane receptors[11]. Ubiquitylation plays a central role in protein degradation through the ubiquitin-proteasome system (UPS)[12]. Decline of UPS activity is an early event during brain aging[13]. Ubiquitin signaling is also required for the turnover of organelles, e.g., mitochondria via mitophagy and endoplasmic reticulum (ER) via ER-phagy[14,15], and it can modulate synaptic activity and plasticity[16,17].

Proteins carrying PTMs, including acetylation, phosphorylation, and ubiquitylation, have been found in protein aggregates in samples from patients suffering from different neurodegenerative diseases[18–20]. For instance, hyperphosphorylation and ubiquitylation of the microtubule-associated protein Tau (MAPT) are characteristic of Alzheimer's disease[18,21]. Similarly, the RNA-binding protein TDP-43 is ubiquitylated and hyperphosphorylated when mislocalized to the cytoplasm in frontotemporal dementia (FTD)[22,23], while GFAP acetylation has been associated with amyotrophic lateral sclerosis[24].

A few previous studies investigated proteome-wide changes in PTMs during vertebrate brain aging. Age- and tissue-specific changes

---

[1]Leibniz Institute on Aging—Fritz Lipmann Institute (FLI), Jena, Germany. [2]Present address: Proteomics Research Infrastructure, University of Copenhagen, Copenhagen, Denmark. [3]Present address: Department of Biology, University of Rochester, Rochester, NY, USA. [4]Present address: UNIL—Université de Lausanne, Lausanne, Switzerland. [5]Present address: Genentech Inc., South San Francisco, CA, USA. [6]These authors contributed equally: Antonio Marino, Domenico Di Fraia. ✉e-mail: alessandro.ori@leibniz-fli.de

in cysteine oxidation have been described[25], e.g., in tRNA aminoacylation complexes, and found not to correlate with protein abundance changes. Protein persulfidation, another redox modification, decreased during rat brain aging and in neurodegenerative disorders[26]. Additionally, in rats, phosphorylation data showed that specific phosphosite levels changed in the aging brain due to mis-localized protein kinases[27]. These studies proved that altering PTMs might contribute to the loss of protein homeostasis in aging and age-related neurodegenerative disorders. However, a systematic investigation of major PTMs during physiological brain aging is still lacking.

To fill this knowledge gap, we quantified the effect of aging on protein acetylation, phosphorylation, and ubiquitylation in the mouse brain using mass spectrometry (MS). We found ubiquitylation to be the most affected PTM. Given the conservation of ubiquitin across evolution[28], we asked whether similar alterations could be observed in different species using the short-lived killifish *Nothobranchius furzeri*. We chose killifish because of its spontaneous age-related brain phenotypes that are common to human neurodegeneration, including accumulation of phosphorylated MAPT with aging[29,30]. By combining mouse and killifish data, we were able to define an ubiquitylation aging signature conserved in the brains of these two species. To identify the causes of altered protein ubiquitylation in the aged brain, we used human induced pluripotent stem cell (iPSC)-derived neurons (iNeurons) and showed that more than one-third of the observed age-related changes in ubiquitylation can be attributed to a decline in proteasome activity. Finally, we tested whether a dietary intervention applied to old mice can influence protein ubiquitylation in the brain and reverse some of the effects of aging.

## Results

### Impact of aging on protein post-translational modification

To assess the impact of aging on PTMs in the mouse brain (C57BL/6J, males), we used label-free data-independent acquisition (DIA) MS and analyzed three major PTMs (ubiquitylation, phosphorylation, and acetylation) from the same set of young and old mice (Fig. 1A). For ubiquitylated peptides enrichment, we used lysine di-GLY (K-$\varepsilon$-GG) remnant motif pulldown[31]. This method leads to the enrichment of other modifications like NEDDylation and ISGylation. However, more than 95% of K-$\varepsilon$-GG-modified sites have been shown to come from ubiquitin[32]. From this point forward, we refer to K-$\varepsilon$-GG-modified sites as ubiquitylated. We quantified changes for 10,487, 7031, and 6049 phosphorylation, ubiquitylation, and acetylation sites, respectively. In parallel, we measured differences in total protein and transcript abundance, covering 6453 proteins and 17,994 transcripts. Our datasets recapitulated changes in PTMs that have been previously associated with aging and disease, e.g., MAPT hyperphosphorylation and GFAP acetylation[24,33] (Fig. S1 and Supplementary Data 1).

Among the analyzed PTMs, we observed a striking effect on the percentage of ubiquitylation sites significantly affected by aging (Fig. 1B). These results prompted us to focus on this specific modification. The ubiquitin dataset's principal component analysis (PCA) consistently depicted a clear separation difference between young and old age groups (Fig. 1C). A higher number of ubiquitylation sites were identified in old samples (Fig. S2B), and age-related changes of ubiquitylation were skewed towards positive values (Fig. 1D). These observations are consistent with previous observations describing increased high-molecular weight ubiquitylated conjugates but not free ubiquitin in mouse brain[34,35], which we further validated in our samples via dot-blot analysis (Fig. S2C). Despite the overall trend of increased ubiquitylation with aging, a fraction of sites showed decreased ubiquitylation (Fig. 1D). When assessing which cellular component categories were affected by these changes, GO enrichment analysis revealed that the myelin sheath, mitochondrion, and GTPase complex showed an increase in ubiquitylation, the latter in agreement with a previous report[36] (Fig. 1E). In contrast, proteins belonging to the synaptic compartment were enriched among those showing decreased ubiquitylation with aging (Fig. 1E). We did not find these changes reflected by proteome and transcriptome datasets that, instead, highlighted a prominent inflammation signature (Fig. S3), a well-described hallmark of aging[37].

Next, we asked whether changes in ubiquitylated peptide abundance could result from underlying alterations of total protein abundance. Although changes in ubiquitylation were positively correlated with changes in protein abundance ($R = 0.47$, $P = 2.2e-16$), 29% of the altered sites could not be explained by altered protein abundance. This observation indicates that changes in ubiquitylation site occupancy occur in the aging brain (Fig. 1F). Among cases that showed a prominent increase in ubiquitylation independently of protein levels, we found proteins encoded by neurodegeneration-associated genes, e.g., APP, TUBB5, and chaperones/co-chaperones, e.g., DNAJB2 (Fig. 1G). In contrast, several histones, deubiquitinases, and synaptic proteins, including H2AFY, USP33, and CAMK2A, showed reduced ubiquitylation (Fig. 1H). Interestingly, we also identified multiple proteins that contained ubiquitylated sites affected oppositely (Fig. S2D–G), indicating a complex rewiring of protein ubiquitylation in the aging mouse brain.

Since ubiquitin-mediated degradation can regulate protein half-life, we correlated our protein ubiquitylation data with protein half-life[38] and protein half-life changes during aging measured in mouse brains by stable isotope labeling coupled to MS[39]. We found that higher levels of ubiquitylation correlated with an increase in protein half-life with aging, but not with half-life itself (Fig. 1I), indicating a link between changes in ubiquitylation and altered proteome turnover in the aging brain. Finally, we investigated whether comparable age-related changes of protein ubiquitylation could be observed in a different organ. Thus, we repeated total and ubiquitylated proteome measurements in the liver of young and old mice (Fig. S4A and Supplementary Data 2). Similarly to the brain, we detected a clear impact of aging on the K-$\varepsilon$-GG-modified proteome, as indicated by PCA (Fig. S4B). In the liver, we observed a comparable number of ubiquitylated sites that showed either increased or decreased modification (Fig. S4C, D), differently from the brain, where most of the sites showed increased ubiquitylation with aging (Fig. 1D). Among the affected ubiquitylation sites, we found an enrichment of proteins related to the cytoskeleton, endosome, and mitochondria displaying increased modification in old mice (Fig. S4E). We found that ubiquitylation changes independent of protein abundance differences are also present in the liver, but to a lower extent than in the brain (significant ubiquitylation sites not significant at the protein level in liver = 333 vs. brain = 1993, Fig. S4F). When we compared directly the effect of aging in two organs, we found only almost no correlation among the shared modified sites after correction for protein abundance ($R = 0.08$, $P = 0.003$, Fig. S4G), with some notable exceptions, e.g., SQSTM1 (p62) that showed increase ubiquitylation at multiple lysines in both organs (Fig. S4G). Together, these results show that protein ubiquitylation is prominently affected in old mice brains, exceeding other protein modifications, and that the ubiquitylation aging signature is largely organ-specific. Additionally, our results indicate that an accumulation of ubiquitylated proteoforms correlates with changes in protein half-life in the aging brain.

### Conserved ubiquitylation changes in aging mice and killifish

Next, we investigated whether age-related alterations in ubiquitylation are conserved across different species by comparing our mouse dataset with one we previously generated for the short-lived killifish *Nothobranchius furzeri*[40]. We hypothesized that any shared changes could reveal a fundamental ubiquitylation signature linked to vertebrate brain aging. First, we confirmed that, also in killifish, a subset of the differences in ubiquitylation are independent of protein

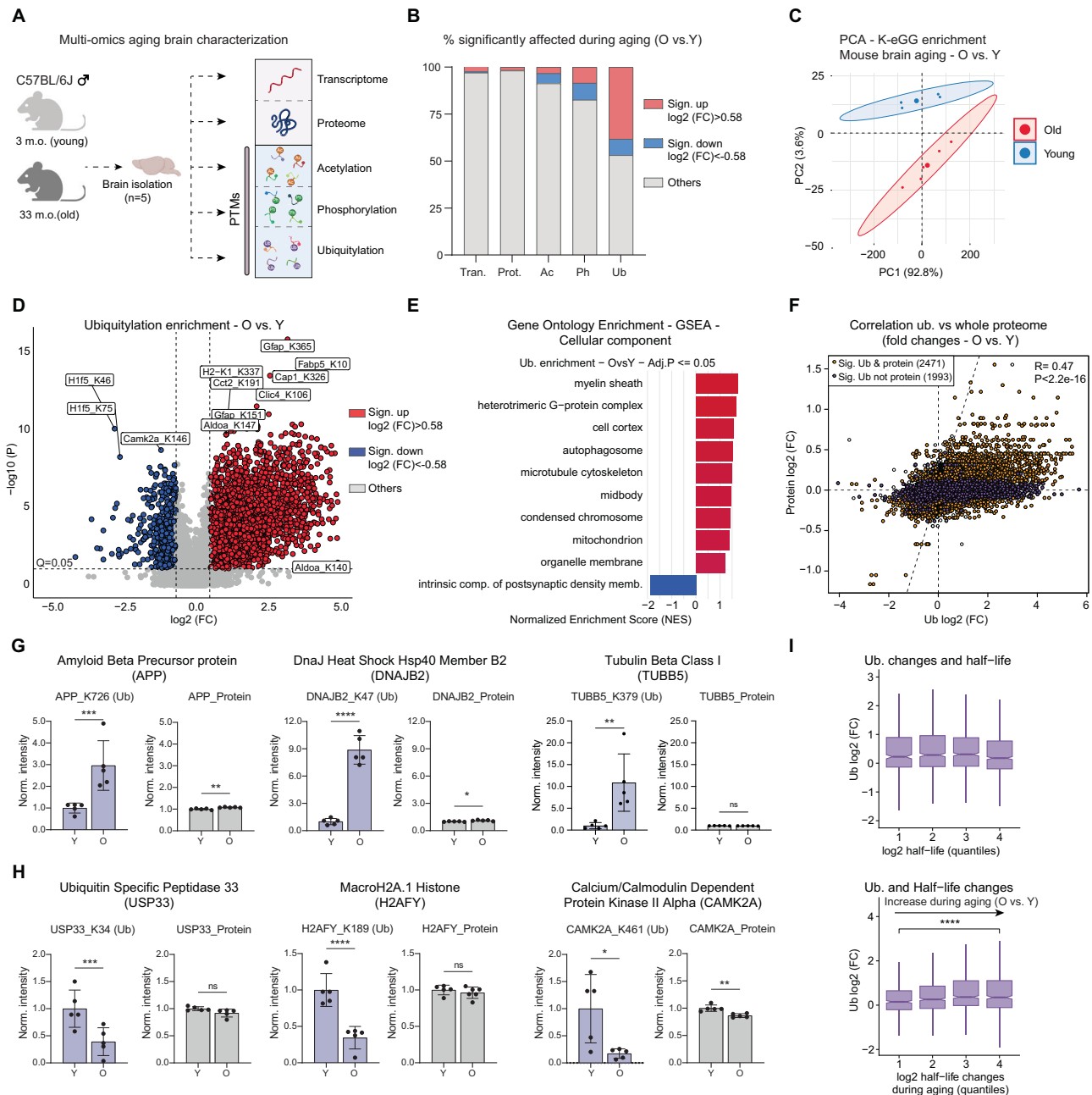

**Fig. 1 | Landscape of protein post-translational modifications in the mouse aging brain. A** Multi-omics approach scheme used to characterize mouse brain aging (3 or 4 vs. 33 months old, $N = 5$, biological replicates, males, C57BL/6J). Created in BioRender. Ori, A. (2025) https://BioRender.com/2947h3v. **B** Percentage of significantly affected transcripts, proteins, or PTMs (Adj.$P < 0.05$ for proteome and transcriptome, $Q < 0.05$ for PTMs; absolute log2 (FC) > 0.58). **C** PCA based on ubiquitylated peptide abundances from mouse brains. Ellipses represent 95% confidence intervals. The percentage of variance explained by each principal component is indicated. **D** Volcano plot for ubiquitin enrichment in mouse brain aging ($N = 5$, biological replicates, $P$ values from Spectronaut differential abundance analysis). **E** Gene set enrichment analysis (GSEA) for ubiquitylated peptides affected by aging based on GO cellular component terms (Adj.$P < 0.05$, weighted Kolmogorov–Smirnov test). **F** Scatterplot illustrating the relationship between fold changes in protein abundance and corresponding ubiquitylated peptide levels with age. Proteins showing significant

age-related changes at both the total protein (Adj. $P < 0.05$ from limma's empirical Bayes moderated $t$-test) and ubiquitylated peptide level ($Q < 0.05$ from Spectronaut differential abundance analysis) are highlighted in orange. Proteins with significant changes in ubiquitylation only (Adj. $P \geq 0.05$; $Q < 0.01$) are shown in purple. Two-sided Pearson's correlation test. **G, H** Examples of proteins showing age-related changes of ubiquitylation ($N = 5$, biological replicates, $Q$ values from Spectronaut for ubiquitylation and Adj.$P$ from limma's empirical Bayes moderated $t$-test for proteome, data shown as averages ± SD). **I** Correlation between ubiquitylation changes and protein half-life[38] (upper panel) or changes in protein half-life[39] during mouse brain aging (bottom panel) (Wilcoxon test). Boxplots show the median, 25th and 75th percentiles (box bounds), and whiskers extending to 1.5 times the Interquartile range; outliers are not shown. *$Q$ or Adj.$P \leq 0.05$; **$Q$ or Adj.$P \leq 0.01$, ***$Q$ or Adj.$P \leq 0.001$, ****$Q$ or Adj.$P \leq 0.0001$. Source data are provided as a Source Data file. Specific $P$ values are listed in Supplementary Data 8. Related to Figs. S1–S4, and Supplementary Data 1 and 2.

abundance (Fig. S5A). Next, we aligned mouse and killifish ubiquitylated sites and compared age-related ubiquitylation changes in the two species upon correction for underlying changes in protein abundance (Supplementary Data 3). Of the 1157 sites we could align between

mouse and killifish, 48% changed consistently or showed a consistent trend when relaxing filtering criteria ($P < 0.25$ in both datasets, Fig. 2A).

Additionally, we examined the correlation between age-related ubiquitylation and proteome differences to determine

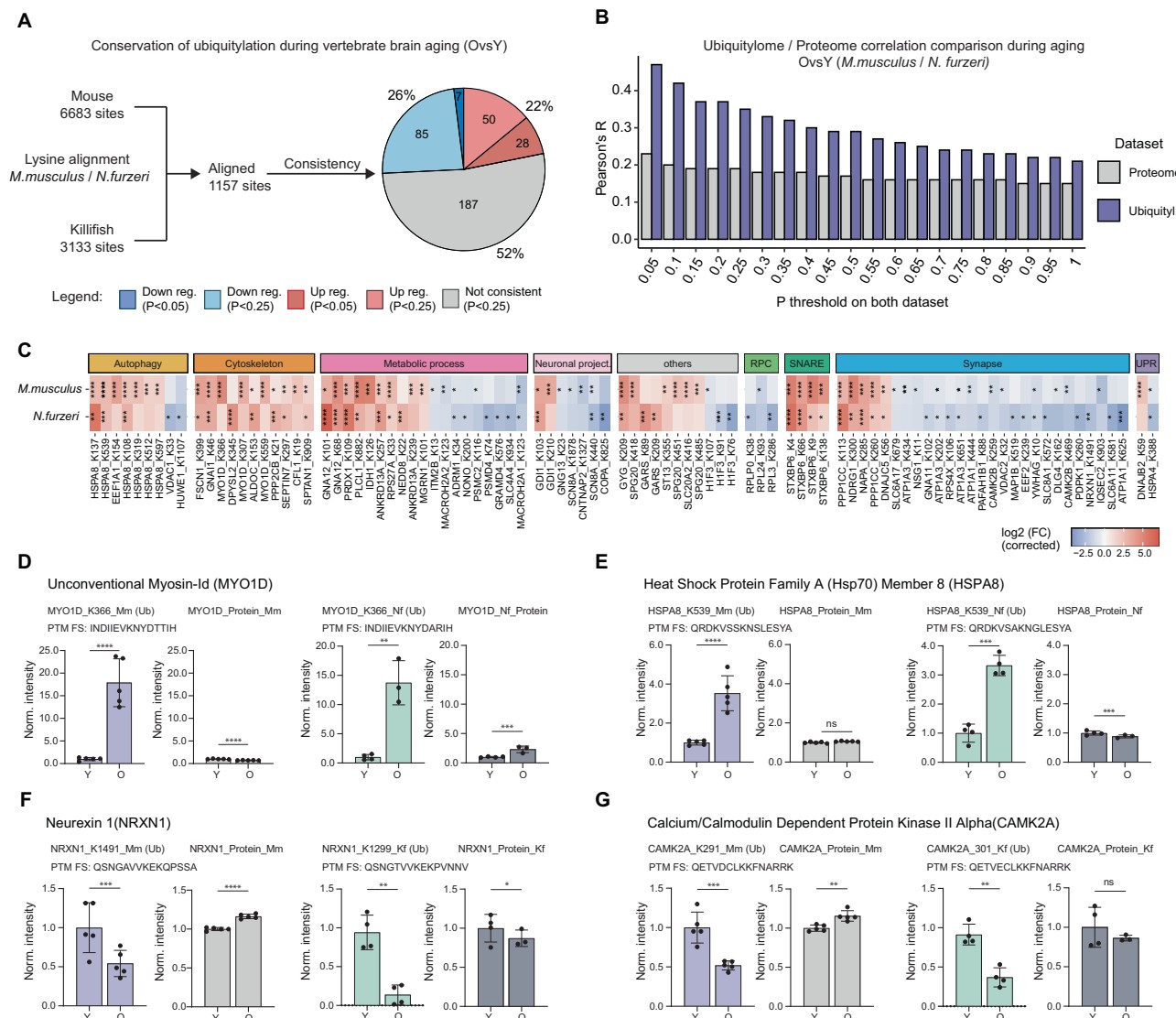

**Fig. 2 | Conservation of age-related ubiquitylation changes in killifish and mouse. A** Mapping of ubiquitylation sites between *M. musculus* and *N. furzeri*. The percentage of consistently up (red and light red for *P* < 0.05 or *P* < 0.25, respectively, in both datasets), down (blue and light blue for *P* < 0.05 or *P* < 0.25, respectively, on both datasets), and not consistent (gray; *P* < 0.25) regulated sites are shown in the pie plot (*N* = 5 for mouse; *N* = 4 for young, and *N* = 3 for old killifish; biological replicates). **B** Barplot showing correlation between age-related changes in ubiquitylated peptides (purple bars) and protein abundance (gray bars) in mouse and killifish. Two-sided Pearson's correlation test. On the y-axis, different P thresholds are applied. **C** Heatmap of age-affected ubiquitylated sites conserved in

mouse and killifish. Lysine position refers to the mouse protein sequence. Fold changes have been corrected for protein abundance (*P* < 0.05 in at least one species). **D–G** Examples of peptides that show age-related changes of ubiquitylation independently of protein abundance, both in mouse and killifish. The flanking sequence (FS) of the modified peptide residues is indicated under the protein name (*N* = 5 for mouse, *N* = 4 for young, and *N* = 3 for old killifish, biological replicates. *Q* values from Spectronaut differential abundance analysis, data shown as averages ± SD). *Q/Adj.P* ≤ 0.05; **Q/Adj.P* ≤ 0.01, ***Q/Adj.P* ≤ 0.001, ****Q/Adj.P* ≤ 0.0001. Source data are provided as a Source Data file. Related to Fig. S5 and Supplementary Data 3.

their conservation across the two species. We found a higher concordance between ubiquitylation changes (expressed as Pearson's correlation between log₂ fold changes) than their respective proteomes, independently of the significance cut-off used (Figs. 2B and S5B, C). When we analyzed which categories of proteins were consistently affected, we identified a subset of proteins related to synapse, autophagy, cytoskeleton, and metabolism (Fig. 2C).

Furthermore, we observed that some of these proteins showed more than one conserved affected site, e.g., MYO1D, HSPA8, and CAMK2A (Fig. 2D–G). MYO1D has been involved in autophagy regulation[41], while HSPA8 is a chaperone that has been reported to prevent tau fibril elongation[42]. CAMK2A, which shows decreased ubiquitylation in old brains, is an essential synaptic plasticity regulator[11].

The robust and conserved effects observed for these proteins suggest them as potential ubiquitylation biomarkers of brain aging.

## Impact of proteasome and lysosome acidification inhibitors on the ubiquitylated proteome of iNeurons

Having identified a protein ubiquitylation aging signature in the brain, we aimed to identify its potential molecular causes. Previous works defined reduced proteasome activity as a hallmark of brain aging[13,35,43–45], a finding that we confirmed in our cohort of mouse samples (Fig. S6A). This prompted us to assess the contribution of reduced proteasome activity to the perturbation of protein ubiquitylation in a relevant model system. We chose to utilize human iPSC-derived neurons (iNeurons)[46] as they have been well-established as a reference in vitro model for human age-associated neurodegenerative

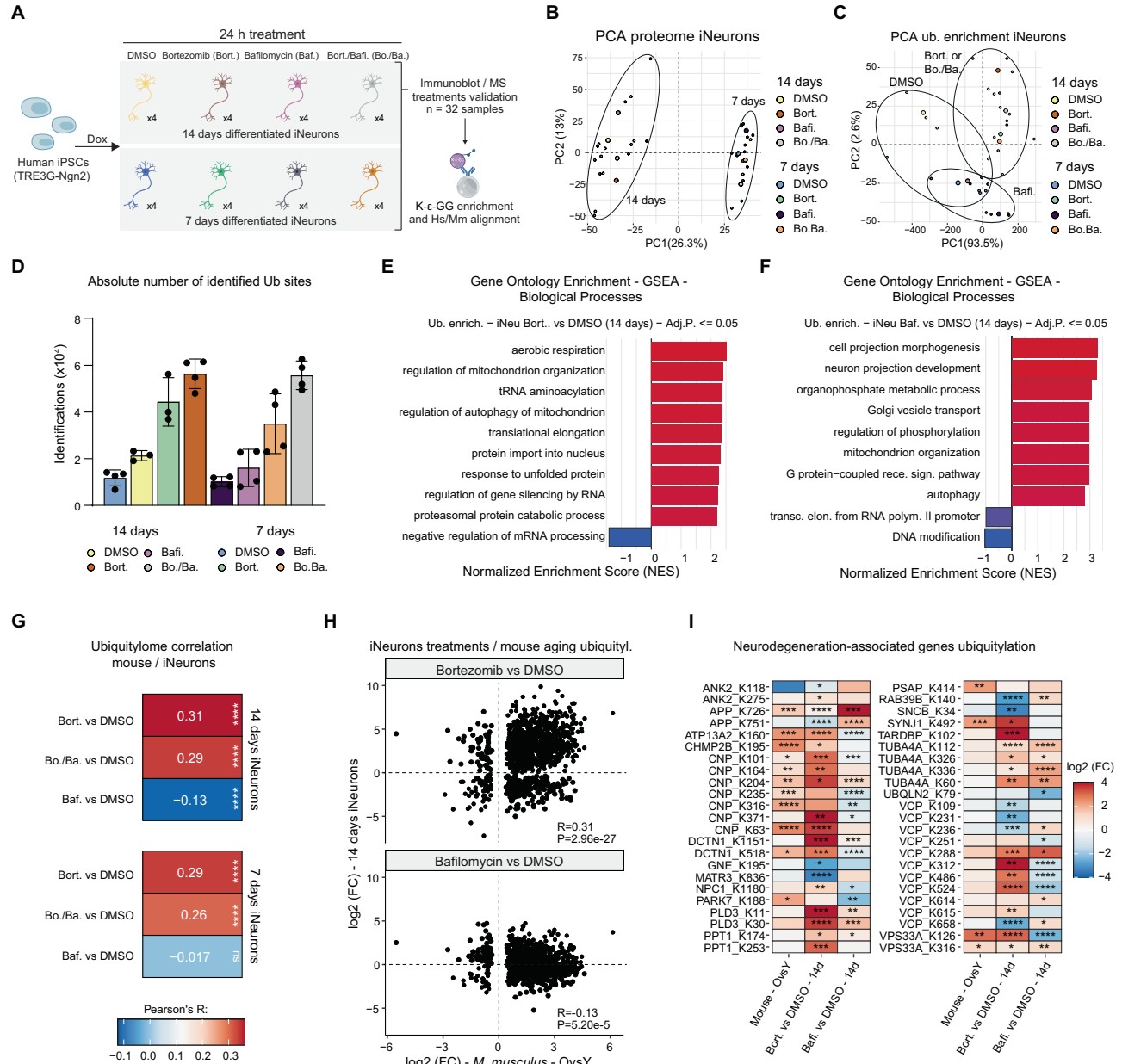

**Fig. 3 | Impact of proteasome inhibition and lysosome acidification impairment on protein ubiquitylation in iNeurons. A** Experimental scheme for iNeurons drug treatment and characterization ($N = 4$, biological replicates from two independent differentiation batches). Created in BioRender. Ori, A. (2025) https://BioRender. com/2947h3v. **B** PCA based on proteome data from iNeurons. The ellipses (drawn manually) highlight the differentiation between day groups. The percentage of variance explained by each principal component is indicated. **C** PCA based on ubiquitylated peptide abundances from iNeurons. The ellipses highlight the drug treatments. The percentage of variance explained by each principal component is indicated. **D** Barplots showing the number of identified ubiquitylated sites in different sample groups ($N = 4$ except $N = 3$ for 14 days DMSO and 7 days bortezomib, biological replicates, data shown as averages ± SD). Gene set enrichment analysis (GSEA) for ubiquitylated peptides affected by bortezomib (**E**) or bafilomycin (**F**) in 14-day iNeurons based on GO cellular component terms (Adj.$P < 0.05$, weighted

Kolmogorov–Smirnov test). **G** Correlation between changes of ubiquitylation observed during mouse brain aging and 14-day iNeurons treated with bortezomib or bafilomycin ($P < 0.05$ in both datasets, two-sided Pearson's correlation test). **H** Scatterplot comparing ubiquitylation changes observed in mouse brain aging and 14-day iNeurons treated with bortezomib (upper panel) or bafilomycin (lower panel) ($P < 0.05$ in both datasets, two-sided Pearson's correlation test). **I** Heatmap of ubiquitylation changes occurring in neurodegeneration-associated proteins. Sites significant ($P < 0.05$, Spectronaut differential abundance test) in at least one of the compared datasets are shown. The lysine numbering refers to the mouse protein sequence. In this figure, mouse ubiquitylation changes were corrected for protein abundance changes. *$P \le 0.05$; **$P \le 0.01$, ***$P \le 0.001$, ****$P \le 0.0001$. Source data are provided as a Source Data file. Specific $P$ values are listed in Supplementary Data 8. Related to Figs. S6–S8, and Supplementary Data 4 and 5.

disorders[47]. Additionally, iNeurons have been effectively used in proteome-wide investigations of protein ubiquitylation[48,49]. Thus, we impaired proteasome activity in 7 days and 14 days post-differentiation iNeurons using bortezomib for 24 h. In parallel, we assessed the contribution of the lysosome-autophagy pathway, as reduced macro-autophagy has been reported in the aging murine brain[50], with several

genes involved in this pathway linked to neurodegeneration in humans[51]. Thus, we also analyzed iNeurons treated with the lysosomal V-ATPase inhibitor bafilomycin, which causes reduced acidification of lysosomes and consequent inhibition of lysosomal proteases, and a third group treated with both bortezomib and bafilomycin (Fig. 3A). No overt cellular toxicity was observed after 24 h of treatment.

However, we noted thinner neuronal projections in bafilomycin-treated neurons (Fig. S6B). We validated the treatments using immunoblot, showing the expected increase of K48-linked ubiquitin chains upon proteasome inhibition (Fig. S6C). We also confirmed an increased LC3B-II/LC3B-I ratio following bafilomycin treatment (Fig. S6C). Next, we used MS to quantify changes in protein abundance and K-ε-GG enriched peptides (Supplementary Data 4). PCA of proteome data highlighted differentiation day as the most prominent signature, although the effect of the different treatments could be observed (Fig. 3B).

Conversely, data from K-ε-GG enriched peptides highlighted a more substantial impact of drug treatments over differentiation days (Fig. 3C). As expected, we noted a robust increase in the number of identified ubiquitylated peptides upon bortezomib treatment, while the number of total proteins remained similar (Figs. 3D and S6D). GO enrichment analysis performed using proteins that displayed changes in ubiquitylation showed that the drug treatments affected different cellular compartments. Bortezomib increased the modification of the mitochondrial and unfolded protein response-related proteins and caused a decrease in the ubiquitylation of factors involved in mRNA maturation (Fig. 3E). Instead, bafilomycin treatment predominantly affected proteins related to neuron projection and Golgi vesicle transport (Fig. 3F). When comparing ubiquitylation and protein abundance changes, we noted a lower correlation than the one we observed in brain tissue (Fig. S7). This difference might be due to the acute treatment modality performed in iNeurons, which renders the contribution of protein abundance changes to ubiquitylation negligible (Fig. S8). Hence, we decided not to apply any additional correction for protein abundance on this particular dataset. To better understand the correlation between the signatures induced by the two drug treatments on cultured neurons and the protein modifications observed in vivo during aging, we aligned human lysine residues to the corresponding mouse and killifish orthologs. We could map 57% of the ubiquitylated lysines for mice (3961 sites) and 45% for killifish (1408 sites) across all iNeurons treatment conditions (Supplementary Data 5).

When we correlated the significantly (P < 0.05) affected sites between iNeurons and mice, we observed that proteasome inhibition, but not lysosomal acidification impairment, could recapitulate a substantial portion of the age-related ubiquitylation changes (Fig. 3G). The majority of the ubiquitylation sites affected by age in mice (~60% of the affected sites, P < 0.05) showed congruent changes in response to the proteasome inhibition treatment (iNeurons, 14 Days). Interestingly, among the shared alterations, SQSTM1 (p62), which is involved in the formation and autophagic degradation of cytoplasmic ubiquitin-containing inclusions[52], showed a prominent increase in ubiquitylation in mouse brain aging and in the bortezomib-treated 14-day iNeurons (Fig. S6E). Although to a lower extent, the same trends were observed for killifish both in terms of correlation (Fig. S6F, G) and percentage of affected sites (~42% of the affected sites, P < 0.05). These results show that partial proteasome inhibition can recapitulate a significant portion of the ubiquitylation changes occurring in vivo during aging, showing a higher correlation and percentage of ubiquitylated sites consistently regulated compared to lysosomal acidification impairment (Fig. 3H).

Next, we used the correlation between aging and the iNeurons dataset to identify groups of proteins whose ubiquitylation depends on or does not depend on proteasome inhibition (Fig. S6H). Gene ontology overrepresentation analysis (ORA) revealed enrichment for cytoskeleton- and chaperone-related proteins among those that exhibited a proteasome- and age-dependent increase in ubiquitylation. Conversely, nucleosome-related proteins were enriched for decreased modification levels in aging and upon proteasome inhibition. We found multiple synaptic and mitochondrial proteins among cases that showed opposite modification patterns. Synaptic proteins exhibited a largely consistent reduction of ubiquitylation in the aging brain, while mitochondrial proteins showed a predominant increase. The same effects were not replicated upon proteasome inhibition (Fig. S6H), suggesting that the effect of aging on the modification of these proteins might depend on other mechanisms.

Finally, as ubiquitylation is a marker of several neurodegenerative diseases, we focused on a subset of proteins genetically linked to neurodegeneration in humans and asked whether their modification state is affected by aging in a proteasome-dependent manner. We mapped multiple ubiquitylation sites on neurodegeneration-associated proteins that consistently changed in mouse brain aging and following bortezomib treatment in 14-day iNeurons (Fig. 3I, Supplementary Data 5). Two proteins, valosin-containing protein (VCP) and 2′,3′-cyclic nucleotide 3′ phosphodiesterase (CNP), showed multiple sites affected by increased ubiquitylation. VCP's main function is to unfold proteins for degradation in the ER, but it also takes part in stress granule clearance and genome stability[53]. It co-localizes with protein aggregates in neurons from patients and mutated protein-expressing cells and is associated with both FTD and inclusion body myopathy[54,55]. CNP is crucial for myelination, creating channels within the myelin sheaths that are essential for proper axon signaling, and its deficiency leads to hypomyelinating leukodystrophy[56], though its link to ubiquitylation is unclear. These findings highlight that age-associated proteostasis impairment, e.g., decreased proteasome activity, can trigger ubiquitylation changes of proteins encoded by neurodegeneration-associated genes even in the absence of mutations.

## Impact of aging and proteasome inhibition on ubiquitin-chain linkages

A critical aspect of protein ubiquitylation is the formation of diverse ubiquitin-chain linkages that mediate distinct cellular signals. While the primary function of K48-linked chains is to target proteins for proteasomal degradation, others, including K6, K27, K33, and K63 linkages, are mainly associated with non-proteasomal pathways such as DNA-damage response, innate immunity, and autophagy regulation, among others[57–61]. Therefore, investigating changes in ubiquitin-chain linkages might shed light on specific molecular mechanisms beyond proteostasis that are altered in aging. To address this, we first examined mouse DIA K-ε-GG enriched data and found that four linkages (K6, K11, K27, and K33) increased significantly during brain aging (Fig. S2H). To validate these findings, we employed targeted proteomics measurements based on parallel reaction monitoring with absolutely quantified isotopically labeled spike-in reference peptides (AQUA-PRM)[62,63](Fig. 4A and Supplementary Data 6). This method enabled us to quantify all linkages, except K29, providing precise measurements of age-related changes across different ubiquitin linkages. Additionally, we applied AQUA-PRM in iNeurons treated with either bortezomib and/or bafilomycin to evaluate the effect of these treatments on the ubiquitin-chain linkages and compare them to brain aging.

First, the AQUA-PRM data confirmed that ubiquitin chains represent only 2–5% of the total ubiquitin pool in murine tissues, similar to previous findings[64], while this proportion increased to 25% in iNeurons (Fig. 4B). While there was no difference in the ratio between total and polyubiquitin pools in young and old mice, we observed a significant increase in polyubiquitin in iNeurons treated with bortezomib (Fig. 4B). Second, we confirmed the increases in total ubiquitin and all lysine linkages observed in the DIA data, except for K27 (Fig. 4C). Notably, similar increases in total ubiquitin, along with K11, K48, and K63 chains, were observed in iNeurons following proteasome inhibition by bortezomib, either alone or combined with bafilomycin (K6 chains were below the limit of detection in iNeurons).

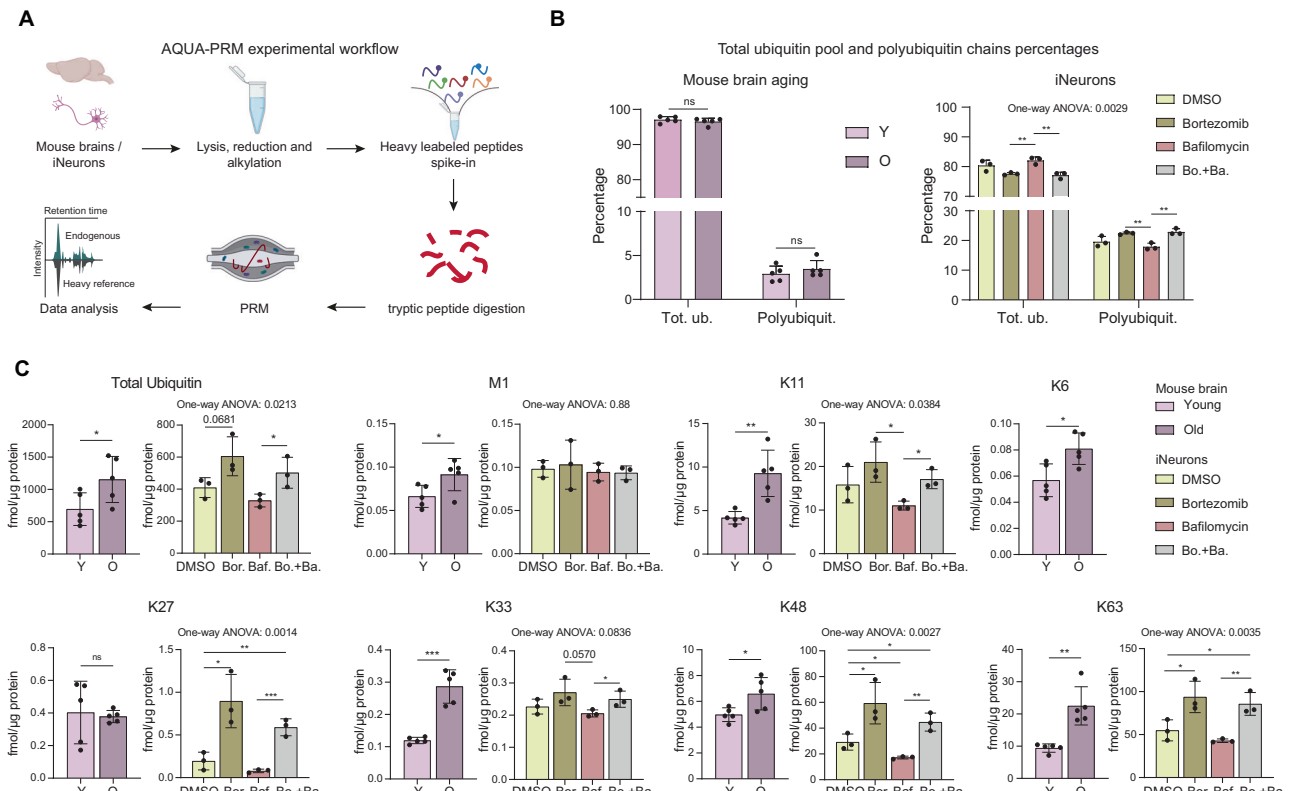

**Fig. 4 | Quantification of ubiquitin-chain linkages in mouse brain and iNeurons.** **A** Experimental scheme for absolute quantification of ubiquitin chains and the total ubiquitin pool ($N = 5$ for mouse, $N = 3$ for iNeurons, biological replicates). Created in BioRender. Ori, A. (2025) https://BioRender.com/2947h3v. **B** Percentage of total and polyubiquitin in mouse brain aging and in iNeurons upon drug treatments measured via AQUA-PRM, * refers to unpaired *t*-test, data shown as averages ± SD. **C** Barplot of total ubiquitin, linear (M1), and branched lysine chains absolute quantification in young and old mice brain and in iNeurons ($N = 5$ for mouse, $N = 3$ for iNeurons, biological replicates, * refers to unpaired *t*-test, One-way ANOVA results are indicated above the barplots, data shown as averages ± SD. *$P \leq 0.05$; **$P \leq 0.01$, ***$P \leq 0.001$, ****$P \leq 0.0001$. Source data are provided as a Source Data file. Specific *P* values are listed in Supplementary Data 8. Related to Supplementary Data 6.

Although the patterns observed in the aging and iNeurons datasets were largely consistent, notable differences emerged. For instance, a pronounced increase in K27 chains was observed in proteasome-impaired iNeurons, absent in aged mouse brains. Conversely, old brains showed a prominent increase in K33 chains observed in iNeurons only when comparing bafilomycin-treated vs. bafilomycin and bortezomib-treated cells (Fig. 4C). These findings indicate that while most of the effects of aging on ubiquitin chains can be mimicked in iNeurons by the proteasome or lysosome acidification impairments, other mechanisms may additionally contribute to specific alterations, e.g., K33 chains, in old brains.

## Impact of dietary intervention on the brain ubiquitylated proteome

Finally, we wanted to investigate whether protein ubiquitylation in the brain can be modified by interventions that affect aging. We focused on dietary interventions that restrict calorie intake since they have been shown to promote cognitive functions and reverse brain aging signatures in various organisms, including rodents and humans[65–68]. To explore whether changes in diet can have lasting effects on protein ubiquitylation in the brain, we subjected 26-month-old male mice to a 30% dietary restriction (DR) for 4 weeks, followed by *ad libitum* (AL) refeeding (RF) for 7 days (Fig. 5A). The RF phase was included to capture persistent changes in brain ubiquitylation following a DR period and to minimize the confounding effects of stress responses and body weight loss (Fig. 5A). Thus, we generated ubiquitylated and whole proteome data from the brains of mice that underwent DR + RF (from here on referred to as RF) and AL fed control mice (Supplementary Data 7). The

proteome of RF mice showed minimal changes (<1% of the quantified proteins, Fig. 5B–D) except for a few proteins known to be affected by DR, such as arginase-1 (ARG1) and carbamoyl phosphate synthetase I (CPS1)[69] (Fig. S9A). In contrast, we detected a pronounced effect of RF on the ubiquitylated proteome (Fig. 5B), with 37% of the quantified ubiquitylated sites being significantly affected (Fig. 5C, D). GO enrichment analysis revealed a significant increase in ubiquitylation of proteins associated with the myelin sheath, lysosomes, mitochondria, and synapses following RF (Fig. S9B). Most of these ubiquitylation changes occurred independently of protein abundance ($R = 0.12$, $P = 4.11e-14$) (Fig. S9C).

Next, we investigated whether age and dietary intervention influence the ubiquitylation of the same protein targets and found that 39% of the cross-quantified sites were significantly affected by both aging and RF (hypergeometric test $P < 2.2e-16$, Fig. 5E). Focusing on the sites affected in at least one of the comparisons, we correlated ubiquitylome changes induced by aging and RF. This analysis revealed a modest but statistically significant positive correlation ($R = 0.15$, $P = 2.9e-19$) between the effect of RF and aging (Fig. 5F). Notably, 1016 ubiquitylation sites showed concordant regulation, indicating that RF may exacerbate some of the aging effects on the ubiquitylated proteome. However, ~23% of the ubiquitylation sites displayed an opposite regulation between aging and RF. To understand how different cellular components are affected by dietary intervention and aging, we compared the enrichment of GO terms among significantly regulated ubiquitylation sites between the two experiments (Fig. 5G). We observed that ubiquitylation of protein components of the myelin sheath, lysosome, and proteasome was exacerbated by the dietary

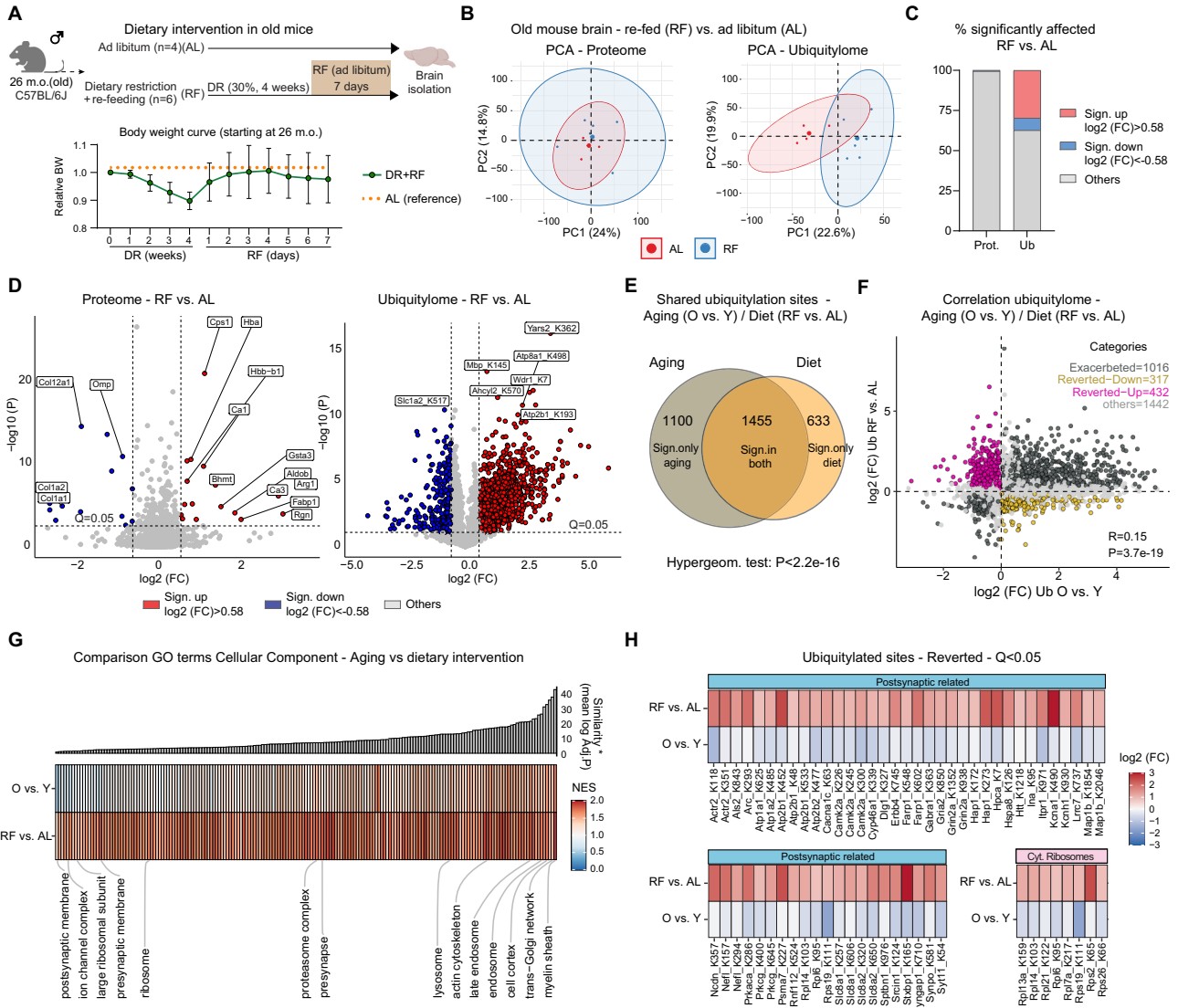

**Fig. 5 | Impact of dietary intervention on the brain proteome and ubiquitylome of old mice. A** Scheme of dietary intervention applied to old mice (*N* = 4 for *ad libitum* fed mice, *N* = 6 for re-fed mice, biological replicates, males, C57BL/6J, 26 months old, data shown as averages ± SD). Created in BioRender. Ori, A. (2025) https://BioRender.com/2947h3v. **B** PCA based on proteome and ubiquitylome data from *ad libitum* (AL) and re-fed mice (RF). Ellipses represent 95% confidence intervals. The percentage of variance explained by each principal component is indicated. **C** Percentage of significantly affected proteins, or ubiquitylated peptides (*Q* < 0.05; absolute log2 fold change (FC) > 0.58). **D** Volcano plot for protein abundance and ubiquitylated peptide changes in RF vs. AL mice (*N* = 4 for AL mice, *N* = 6 for RF mice, biological replicates). **E** Venn diagram displaying overlap between ubiquitylation significantly (*Q* < 0.05) affected by aging and dietary intervention (508 sites not significant in both datasets are not displayed, Hypergeometric test). **F** Scatterplot illustrating the relationship between ubiquitylated peptide changes observed with aging and those following dietary intervention in old mice. Only peptides with significant changes (*Q* < 0.05) in at least one dataset are shown. Peptides with congruent and significant changes in both datasets (*Q* < 0.05) are labeled as "Exacerbated" (dark gray). "Reverted-down" (yellow) denotes peptides that significantly decreased upon dietary intervention (*Q* < 0.05 in RF data) in the opposite direction to the age-related increase. "Reverted-up" (purple) indicates peptides that significantly increased upon intervention in the opposite direction to the age-related decrease. Peptides with consistent directional changes but significant in only one dataset are shown in light gray. Two-sided Pearson's correlation test. **G** Heatmap comparing the normalized enrichment scores (NES) from gene set enrichment analyses (GSEA) performed on age-related and diet-induced changes of ubiquitylation. GO terms are sorted by their scaled similarity (1/Euclidean distance between NES), multiplied by the average −log10 (Adj.*P*) across the two datasets, weighted Kolmogorov–Smirnov test. **H** Heatmap highlighting age-related changes in ubiquitylation that are reverted by dietary intervention (*Q* < 0.05 in response to dietary intervention and of opposite log2 (FC) in response to aging). Source data are provided as a Source Data file. Related to Fig. S9 and Supplementary Data 7.

intervention (Fig. S9D), while ubiquitylation of postsynaptic membrane proteins, large ribosomal subunit, and others was reverted (Figs. 5H and S9E).

Taken together, these results indicate that a dietary intervention comprising a period of DR followed by RF has a lasting impact on brain protein ubiquitylation. Aging and diet affect the ubiquitylation of an overlapping set of proteins, with some sites showing reversed and others exacerbated modification, highlighting the complex effects of dietary intervention on protein homeostasis in the old brain.

## Discussion

In this work, we investigated the impact of aging on three commonly studied PTMs in the vertebrate brain. Our findings reveal that changes in protein ubiquitylation far exceed the signatures that can be detected from the same sample at the level of transcriptome, proteome, or other analyzed PTMs. Many ubiquitylation changes are independent of protein abundance, and if consistent, they show considerably more pronounced effect sizes. These differences cannot be attributed to technical artifacts related to peptide versus protein quantification, as

other PTMs that undergo similar experimental (peptide enrichment) and analytical procedures do not exhibit such pronounced effects. Since ubiquitylation is a main regulator of protein degradation and homeostasis, one might expect that changes in this PTM would reflect equal, if not greater, effects at the protein level. However, in line with previous in vitro observations from cultured cells[32], our study shows this is not always the case. A possible explanation for the apparent paradox of changes of ubiquitylation that do not manifest in comparable protein level changes might reside in the fact that ubiquitylated proteoforms represent only a small percentage of the total protein pool[70], making their changes in abundance difficult to detect in global proteome data. The accumulation of ubiquitylated proteoforms might result from the sequestration of misfolded proteins in aggregates that can protect cells from further proteostasis impairment. Such a mechanism has been proposed by several studies[71–73]. Additionally, a subset of the ubiquitylation sites might not represent PTMs directly involved in protein degradation. Nevertheless, we observed a significant correlation between ubiquitylation and protein half-life changes[39] during mouse brain aging (Fig. 1I). This observation, together with the concomitant reduction of proteasome activity, points to a mechanistic link between altered protein turnover and accumulation of ubiquitylated proteoforms.

We describe a major remodeling of the protein ubiquitylation landscape, which is conserved in two vertebrate species (mouse and killifish) and also occurs in the liver. The remodeling of the protein ubiquitylation landscape, defined as hundreds of sites showing either increased or decreased modification with aging, is in line with previous data from *C. elegans*[74], and it demonstrates the conservation of this aspect of age-related proteostasis impairment in vertebrate species. Notably, an independent study reported that >90% of the proteins that become metastable in aging nematodes undergo changes of ubiquitylation[75], further underscoring a correlation between PTM and age-related loss of proteostasis. Different from our brain data that point to a bias towards increased ubiquitylation in old brains, the *C. elegans* data shows a global decrease in ubiquitylation with aging. This difference might arise from cell type-specific (whole organism vs. brain) or species-specific effects (nematodes vs. vertebrates). In support of a brain-specific accumulation of ubiquitylated proteins, we noted a less pronounced increase in ubiquitylation in the aging mouse liver.

In the brain, most of the quantified ubiquitylation sites showed increased modification, consistent with a global increase of protein ubiquitylation during brain aging[34,35,76]. Among proteins showing the strongest increase in ubiquitylation both in mouse and killifish brains, we found several chaperons and co-chaperons, e.g., DNAJB2, DNAJB6, and HSPA8. HSPA8 has been reported to interact with tau fibrils[42], while DNAJB1 and DNAJB6 are part of the DNAJ family, a group of co-chaperons that play a crucial role in regulating the activity of HSPA1 and assist in its proper functioning[77]. These proteins might become hyper-ubiquitylated upon sequestration in protein aggregates[78].

Although most ubiquitylation sites show increased modification with aging, a subset decreases. Particularly, synaptic proteins were affected by a general decrease in ubiquitylation (Fig. 1E), and previous reports showed that synaptic transmission is controlled by UPS[12]. Interestingly, a substantial decrease in ubiquitylation was previously observed following the excitation of rat synaptosomes and was not reverted when blocking the proteasome, suggesting that deubiquitylases might regulate this process[79]. Along this line, both overexpression of ubiquitin and depletion of deubiquitylases, e.g., USP14 and UCHL1, have been shown to cause structural alterations of the neuromuscular junction and synaptic transmission[16,80,81]. Our data suggests that altered ubiquitylation of the synaptic compartment might play a role in how synaptic function and plasticity are altered in aging. Intriguingly, we found some of the age-related alterations of synaptic protein ubiquitylation to be reversed by a dietary intervention based on DR followed by RF. It is tempting to speculate that some of the beneficial effects of this type of intervention on cognitive function might be mechanistically linked to a restoration of protein ubiquitylation dynamics in the synaptic compartment.

Another notable instance of decreased ubiquitylation in our dataset involves USP33, a deubiquitylase localized to the mitochondrial outer membrane and known to interact with Parkin, an E3 ubiquitin ligase associated with Parkinson's disease. USP33 depletion has been shown to increase K63-linked ubiquitin conjugates on Parkin, leading to its stabilization and enhanced mitophagy[82]. Thus, an age-related decline in ubiquitylation could stabilize and activate USP33, potentially modulating Parkin-dependent mitophagy. Altered mitophagy might, in turn, contribute to the elevated ubiquitylation of mitochondrial proteins observed in aged brains (Fig. 1E).

In an attempt to pinpoint causes of altered protein ubiquitylation in the aging brain, we focused on protein clearance pathways, given that their reduced activity has been linked to both aging and neurodegeneration[13,45,50,51,83–86]. We experimentally defined that more than one-third of the age-related changes in protein ubiquitylation are causally linked to decreased proteasome activity. This figure likely represents an underestimation given the limitation of our experimental setup based on a single cell type (iPSC-derived glutamatergic neurons, iNeurons). Our analysis might have missed ubiquitylation sites from proteins expressed by other cell types, or that did not align between mouse and human. An additional potential limitation might derive from the acute nature of our perturbations in vitro (24 h), while aging cells experience in vivo a chronic reduction of proteasome activity.

The effect of proteasome activity perturbation was also assessed for the ubiquitin-chain linkages. This analysis confirmed that the age-related upregulation of K48-linked chains was consistent with proteasome impairment. Interestingly, the K63 and K11 linkages also had a pronounced increase. K11 has been shown to accumulate upon proteasome inhibition in yeast and U2OS human cells[87,88], while K63 might reflect the effort by the cell to enhance protein clearance by lysosomal degradation[89]. The AQUA-PRM analysis also revealed a prominent increase in the non-canonical K33-linked chains that could not be recapitulated by the drug treatment alone in iNeurons. K33 polyubiquitin chain-modified substrates are known to be involved in protein trafficking via F-actin assembly[90] and promoting autophagy by interacting with ubiquitin-binding autophagy receptors, such as LC3B and SQSTM1[91]. Regulation of K33 ubiquitin chains has been linked to the activity of specific deubiquitylase, i.e., TRABID (ZRANB1)[92], which, upon deletion, leads to decreased lifespan in *Drosophila* by modulating the innate immune response[93]. Thus, additional mechanisms like changes in macroautophagy or the activity of deubiquitylases might contribute to the rewiring of the protein ubiquitylation landscape, promoting brain aging in vertebrates, as shown in nematodes[74].

The effects of change in diet on protein ubiquitylation in the brain have not been studied in detail, although experimental evidence linking diet and ubiquitylation exists. In flies, time-restricted feeding led to a decrease in the area of polyubiquitin and SQSTM1 (p62) aggregates in muscles[94]. Similarly, DR reduced the age-related increase in ubiquitylation in the mouse heart[95]. DR has also been shown to reverse the age-related impairment of proteasomal activity in the brain, but not in the liver of mice[43]. Conversely, a prolonged high-fat diet led to an impairment of the UPS, resulting in the accumulation of ubiquitinated proteins and an increased colocalization of ubiquitin and SQSTM1 (p62) in the arcuate nucleus[96]. Here, we have shown that DR induces detectable changes in protein ubiquitylation in the brain that persist for at least 7 days after starting the RF phase. Importantly, the ubiquitylation sites influenced by diet and aging overlap, but the directionality of the regulation differs between sites. We observed a mild positive correlation between the effect of RF and aging, suggesting that this type of dietary intervention might exacerbate the impact of aging on the ubiquitylated proteome. Some of these alterations might be related to the potential deleterious effects of dietary interventions when applied in old age. For

instance, Hahn and colleagues demonstrated that while late-life DR produced modest survival benefits and limited metabolic remodeling in mice, it also caused an acute increase in mortality when switching from a dietary to an AL diet[97]. Despite this general trend, we also identified notable exceptions where dietary intervention reversed the effect of aging on protein ubiquitylation. These include synaptic (discussed above) and ribosomal proteins, both classes of proteins that showed conserved age-related changes of ubiquitylation in mice and killifish. Given that ribosome dysfunction has been associated with brain aging and neurodegeneration[39,40,98,99] and that ubiquitylation of ribosomal proteins has been linked to different types of stressors[100,101], our data might indicate a potential beneficial effect of dietary intervention on protein synthesis impairment in the aging brain. Recent data have linked RF to the enhancement of protein synthesis in intestinal stem cells via activation of the polyamine pathway[102]. Intriguingly, deficiency in the polyamine pathway has been linked to both Alzheimer's and Parkinson's disease in humans[103,104].

More broadly, we identified several ubiquitylation sites in the brain on proteins genetically linked to human neurodegeneration. Importantly, we found multiple of these sites to be affected by aging (Fig. 3I and Supplementary Data 5). Although it remains unclear whether these PTMs are mechanistically linked to the pathogenic properties of the mutated forms of these proteins or whether the same modifications occur in diseased human tissues, our findings suggest that ubiquitylated proteoforms are generally more susceptible to age-related changes and may represent more robust biomarkers of brain aging and, potentially, neurodegeneration.

## Methods

### Mice
All wild-type mice were C57BL/6J obtained from Janvier Labs (Le Genest-Saint-Isle, France) or internal breeding at FLI. All animals were kept in a specific pathogen-free animal facility with a 12 h light/dark cycle at a temperature of 20 °C ± 2 and humidity of 55% ± 15. Young mice were aged 3 or 4 months, and old mice were aged 33 months. During the experiment, Mice had unlimited access to food (ssniff, Soest, Germany). For the proteomic and transcriptomic analysis, only male mice were used, and in all the figures, N refers to biological replicates. Mice were euthanized with $CO_2$, and organs were isolated, washed in PBS, weighed, and immediately snap-frozen in liquid nitrogen before storage at −80 °C. All experiments were carried out according to the guidelines from Directive 2010/63/EU of the European Parliament on the protection of animals used for scientific purposes. The protocols of animal maintenance and euthanasia were approved by the local authorities for animal welfare in the State of Thuringia, Germany (Reg.-Nr. O_AO_18-20, O_JvM_18-20 and FLI-20-003).

### Dietary restriction and re-feeding mice experiment
Mice were separated into single cages 2 weeks before the dietary treatment. Food weight and body weight were measured directly after separating the animals and before the dietary treatment. The daily food intake per animal was calculated and used for calculating the amount that refers to 70% for every animal. Food was given in the morning, once per day, for 1 month of the DR period. AL animals had unlimited access to food throughout the experiment. Finally, animals of the DR + RF cohort received unlimited access to food for 7 days after the DR period. Experiments were conducted according to protocols approved by the state government of Thuringia, Thüringer Landesamt für Verbraucherschutz authority (license number: FLI-20-003). Only male mice were used, and in all the figures, N refers to biological replicates.

### hiPSC cell line maintenance and differentiation
The protocol used for the maintenance of the WTC11 hiPSC cell line is described in detail by ref. 46. In brief, the Matrigel (Corning, 356231) coating solution was prepared by diluting 100X concentrated Matrigel

Matrix in DMEM/F12 (Gibco, 11320033) medium and coating the tissue culture surface with it. Thawed cells were suspended in Essential 8 (E8) culture medium (CM) (Gibco, A1517001) supplemented with Y-27632 ROCK inhibitor (Abcam, ab120129) and transferred to the Matrigel-coated plate. The culture was maintained by daily E8 medium change without ROCK inhibitor. Cells were split using Accutase solution (Sigma, A6964) and resuspended in prewarmed E8 CM supplemented with ROCK inhibitor before transfer to the Matrigel-coated plate. For long-term storage, cells were frozen in cryopreservation medium (10% DMSO, 20% FBS in E8 medium) and transferred to liquid nitrogen. WTC11 hiPSC Cell Line was a kind gift from the Ward Lab.

### iPSCs differentiation into iNeurons and drug treatments
Cells were collected using Accutase solution (Sigma, A6964) when they reached 70%-80% confluency. After centrifugation, the cell pellet was resuspended and transferred to a Matrigel-coated plate. The induction medium (IM) (DMEM/F12 (Gibco, 11320033) supplemented with N2 supplement (Gibco, 17502048), MEM NEAA (Gibco, 11140050), L-glutamine (Gibco, 25030081) and 2 μg/mL doxycycline (Sigma, D9891)) supplemented with 10 μM Y-27632 ROCK inhibitor (Abcam, ab120129) was then added to the cells, which were incubated at 37 °C overnight. The IM without ROCK inhibitor was exchanged on the second day, and on the third day, cells were collected using Accutase solution and either frozen or re-plated for neuronal maturation. To culture iNeurons, PLO-coating was performed (1 mg/mL Poly-L-ornithine hydrobromide (Sigma-Aldrich, 27278-49-0) in the buffer containing 100 mM boric acid, 25 mM sodium tetraborate, 75 mM sodium chloride and 1 M sodium hydroxide), and freshly dissociated 3-day differentiated iNeurons were resuspended in cortical neuron CM (Neurobasal medium (Gibco, 21103049)) supplemented with B-27 supplement (Gibco, 17504044), 10 ng/mL BDNF (PeproTech, 450-02), 10 ng/mL NT-3 (PeproTech, 450-03) and 1 μg/mL laminin (Sigma-Aldrich, 114956-81-9) supplemented with 2 μg/mL doxycycline and 10 μM Y-27632 ROCK inhibitor. Half-medium changes with CM were performed biweekly to maintain the culture.

The proteasome and lysosome acidification inhibitions were performed in either 7-day or 14-day iNeurons. Proteasome inhibition was achieved by treating cells with bortezomib (Sigma, 5043140001), and lysosome acidification was inhibited using Bafilomycin A1 from S. griseus (Sigma, B1793). Vehicle solution (DMSO) and 20 nM drug solutions were prepared in one-half culture volume of fresh CM. Half of the CM was removed from the cell culture plate directly before treatment and substituted with an equal amount of the corresponding, fresh-prepared drug solution, resulting in a 10 nM final concentration of the drug. Plates were then returned to the incubator and incubated for 24 h. In all the figures, N refers to biological replicates.

### Sample preparation for total proteome and analysis of PTMs
For iNeurons samples, cell pellets were used directly for the lysis step. For snap-frozen brains and livers, samples were thawed and transferred into Precellys® lysing kit tubes (Keramik-kit 1.4/2.8 mm, 2 mL (CKM)) containing PBS supplemented with cOmplete™, Mini, EDTA-free Protease Inhibitor (Roche, 11836170001) and with PhosSTOP™ Phosphatase Inhibitor (Roche, 4906837001). The volume of PBS added was calculated based on an estimated protein content (5% of fresh tissue weight) to reach a 10 μg/μL concentration. Tissues were homogenized twice at 6000 rpm for 30 s using Precellys® 24 Dual (Bertin Instruments, Montigny-le-Bretonneux, France), and the homogenates were transferred to new 2 mL Eppendorf tubes.

Samples corresponding to ~1.5 mg of protein were used as starting material for each biological replicate. Volumes were adjusted using PBS, and samples were lysed by the addition of 4× lysis buffer (8% SDS, 100 mM HEPES, pH 8, not containing NEM). Samples were sonicated twice in a Bioruptor Plus for 10 cycles with 1 min ON and 30 s OFF with high intensity at 20 °C. The lysates were centrifuged at $18,407 \times g$ for

1 min and transferred to new 1.5 mL Eppendorf tubes. Subsequently, samples were reduced using 20 mM DTT (Carl Roth, 6908) for 15 min at 45 °C and alkylated using freshly made 200 mM iodoacetamide (IAA) (Sigma-Aldrich, I1149) for 30 min at room temperature in the dark. An aliquot of each lysate was used for estimating the precise protein quantity using Qubit protein assay, and volumes were adjusted accordingly to have a total protein amount of 1.25 mg (Thermo Scientific, Q33211). Subsequently, proteins were precipitated using cold acetone, as described in ref. 105, and resuspended in 500 μL of digestion buffer (3 M urea, 100 mM HEPES pH 8.0). Proteins were digested using LysC 1:100 enzyme:proteins ratio for 4 h (Wako sequencing grade, 125-05061) and trypsin 1:100 enzyme:proteins ratio for 16 h (Promega sequencing grade, V5111). The digested proteins were then acidified with 10% (v/v) trifluoroacetic acid. Aliquots corresponding to 20, 200, and 1000 μg of peptides were taken for proteome, phosphopeptides, and ubiquitylated / acetylated peptides enrichment, respectively, and desalted using Waters Oasis® HLB μElution Plate 30 μm (2, 10, and 30 mg, depending on the amount of starting material) following manufacturer instructions. The eluates were dried down using a vacuum concentrator and reconstituted in MS buffer A (5% (v/v) acetonitrile, 0.1% (v/v) formic acid). For PTM enrichment, peptides were further processed as described below. For DIA based analysis of total proteome, samples were transferred to MS vials, diluted to a concentration of 1 μg/μL, and spiked with iRT kit peptides (Biognosys, Ki-3002-2) before analysis by LC-MS/MS.

## Sequential enrichment of ubiquitylated and acetylated peptides

Ubiquitylated and acetylated peptides were sequentially enriched starting from ~1000 μg of dried peptides per replicate. For the enrichment of ubiquitylated peptides, the PTMScan® HS Ubiquitin/ SUMO Remnant Motif (K-ε-GG) kit (Cell Signaling Technology, 59322) was used following the manufacturer's instructions. The K-ε-GG-modified enriched fraction was desalted and concentrated as described above, dissolved in MS buffer A, and spiked with iRT kit peptides before LC-MS/MS analysis.

The flowthrough fractions from the K-ε-GG enrichment were acidified with 10% (v/v) trifluoroacetic acid and desalted using Oasis® HLB μElution Plate 30 μm (30 mg) following the manufacturer's instructions. Acetylated peptides were enriched as described by ref. 106. Briefly, dried peptides were dissolved in 1000 μL of IP buffer (50 mM MOPS pH 7.3, 10 mM KPO₄ pH 7.5, 50 mM NaCl, 2.5 mM Octyl β-D-glucopyranoside) to reach a peptide concentration of 1 μg/μL, followed by sonication in a Bioruptor Plus (5 cycles with 1 min ON and 30 s OFF with high intensity at 20 °C). Agarose beads coupled to an antibody against acetyl-lysine (ImmuneChem Pharmaceuticals Inc., ICP0388-5MG) were washed three times with washing buffer (20 mM MOPS pH 7.4, 10 mM KPO4 pH 7.5, 50 mM NaCl) before incubation with each peptide sample for 1.5 h on a rotating well at 750 rpm (STARLAB Tube roller Mixer RM Multi-1). Samples were transferred into Clearspin filter microtubes (0.22 μm) (Dominique Dutscher SAS, Brumath, 007857ACL) and centrifuged at 4 °C for 1 min at 2000 × *g*. Beads were washed first with IP buffer (three times), then with washing buffer (three times), and finally with 5 mM ammonium bicarbonate (three times). Thereupon, the enriched peptides were eluted first in basic conditions using 50 mM aqueous NH3, then using 0.1% (v/v) trifluoroacetic acid in 10% (v/v) 2-propanol and finally with 0.1% (v/v) trifluoroacetic acid. Elutions were dried down and reconstituted in MS buffer A (5% (v/v) acetonitrile, 0.1% (v/v) formic acid), acidified with 10% (v/v) trifluoroacetic acid, and then desalted with Oasis® HLB μElution Plate 30 μm. Desalted peptides were finally dissolved in MS buffer A, spiked with iRT kit peptides, and analyzed by LC-MS/MS.

## Enrichment of phosphorylated peptides

Desalted peptides corresponding to 200 μg, as described in "Sample preparation for total proteome and analysis of PTMs," were used. The last desalting step was performed using 50 μL of 80% ACN and 0.1% TFA buffer solution. Before phosphopeptide enrichment, samples were filled up to 210 μL using 80% ACN and 0.1% TFA buffer solution. Phosphorylated peptides were enriched using Fe (III)-NTA cartridges (Agilent Technologies, G5496-60085) in an automated fashion using the standard protocol from the AssayMAP Bravo Platform (Agilent Technologies). In short, Fe (III)-NTA cartridges were first primed with 100 μL of priming buffer (100% ACN, 0.1% TFA) and equilibrated with 50 μL of buffer solution (80% ACN, 0.1% TFA). After loading the samples into the cartridge, the cartridges were washed with an OASIS elution buffer, while the syringes were washed with a priming buffer (100% ACN, 0.1% TFA). The phosphopeptides were eluted directly with 25 μL of 1% ammonia into 25 μL of 10% FA. Samples were dried with a speed vacuum centrifuge and stored at −20 °C until LC-MS/MS analysis.

## Data-independent acquisition mass spectrometry

Peptides were separated in trap/elute mode using the nanoAcquity MClass Ultra-High Performance Liquid Chromatography system (Waters, Waters Corporation, Milford, MA, USA) equipped with trapping (nanoAcquity Symmetry C18, 5 μm, 180 × 20 mm) and an analytical column (nanoAcquity BEH C18, 1.7 μm, 75 μm × 250 mm). Solvent A was water and 0.1% formic acid, and solvent B was acetonitrile and 0.1% formic acid. 1 μL of the samples (~1 μg on column) was loaded with a constant flow of solvent A at 5 μL/min onto the trapping column. Trapping time was 6 min. Peptides were eluted via the analytical column with a constant flow of 0.3 μL/min. During the elution, the percentage of solvent B increased nonlinearly from 0 to 40% in 120 min. The total run time was 145 min, including equilibration and conditioning. The LC was coupled to an Orbitrap Exploris 480 (Thermo Fisher Scientific, Bremen, Germany) using the Proxeon nanospray source. The peptides were introduced into the mass spectrometer via a Pico-Tip Emitter 360-μm outer diameter × 20-μm inner diameter, 10-μm tip (New Objective), heated at 300 °C, and a spray voltage of 2.2 kV was applied. The capillary temperature was set at 300 °C. The radio frequency ion funnel was set to 30%. For DIA data acquisition, full scan MS spectra with a mass range 350–1650 *m/z* were acquired in profile mode in the Orbitrap with a resolution of 120,000 FWHM. The default charge state was set to 3+. The filling time was set at a maximum of 60 ms with a limitation of $3 \times 10^6$ ions. DIA scans were acquired with 40 mass window segments of differing widths across the MS1 mass range. Higher collisional dissociation fragmentation (stepped normalized collision energy; 25, 27.5, and 30%) was applied, and MS/MS spectra were acquired with a resolution of 30,000 FWHM with a fixed first mass of 200 *m/z* after accumulation of $3 \times 10^6$ ions or after filling time of 35 ms (whichever occurred first). Data were acquired in profile mode. Xcalibur 4.3 (Thermo) and Tune version 2.0 were used to acquire and process the raw data.

## TMT labeling

TMT was used for mouse brain proteome data. The solution containing the resuspended peptides was brought to a pH of 8.5 and a final concentration of 100 mM HEPES (Sigma H3375) prior to labeling. 20 μg of peptides were used for each label reaction. TMT-10plex reagents (Thermo Fisher #90111) were reconstituted in 41 μL of acetonitrile (Biosolve #0001204102BS). TMT labeling was performed in two steps by the addition of 2× of the TMT reagent per mg of peptide (e.g., 40 μg of TMT reagent for 20 μg of peptides). TMT reagents were added to samples at room temperature, followed by incubation in a thermomixer (Eppendorf) under constant shaking at 600 rpm for 30 min. After incubation, a second portion of TMT reagent was added and followed by incubation for another 30 min. After checking the labeling efficiency by MS, equal amounts of samples were pooled (200 μg total), desalted using two wells of a Waters Oasis HLB mElution Plate 30 mm (Waters #186001828BA), and subjected to high pH

fractionation before MS analysis. The labeling of the samples was performed as follows: Young1 (126), Young2 (127_N), Young3 (127_C), Young4 (128_N), Young5 (128_C), Old1 (129_N), Old2 (129_C), Old3 (130_N), Old4(130_C), Old5 (131).

## High pH peptide fractionation

Offline high pH reverse phase fractionation was performed using an Agilent 1260 Infinity HPLC System equipped with a binary pump, degasser, variable wavelength UV detector (set to 220 and 254 nm), peltier-cooled autosampler (set at 10 °C), and a fraction collector. The column used was a Waters XBridge C18 column (3.5 μm, 100 x 1.0 mm, Waters) with a Gemini C18, 4 × 2.0 mm SecurityGuard (Phenomenex) cartridge as a guard column. The solvent system consisted of 20 mM ammonium formate (20 mM formic acid (Biosolve #00069141A8BS), 20 mM (Fluka #9857) pH 10.0) as mobile phase (A) and 100% acetonitrile (Biosolve #0001204102BS) as mobile phase (B). The separation was performed at a mobile phase flow rate of 0.1 mL/min using a non-linear gradient from 95% A to 40% B for 91 min. Forty-eight fractions were collected along with the LC separation and subsequently pooled into 24 fractions. Pooled fractions were dried in a speed vacuum centrifuge and then stored at −80 °C until MS analysis.

## Data acquisition for TMT-labeled samples

For TMT experiments, fractions were resuspended in 20 μL reconstitution buffer (5% (v/v) acetonitrile (Biosolve #0001204102BS), 0.1% (v/v) TFA in water) and 5 μL were injected into the mass spectrometer. Peptides were separated using the nanoAcquity UPLC system (Waters) fitted with a trapping (nanoAcquity Symmetry C18, 5 μm, 180 μm × 20 mm) and an analytical column (nanoAcquity BEH C18, 2.5 μm, 75 μm × 250 mm). The analytical column outlet was coupled directly to an Orbitrap Fusion Lumos (Thermo Fisher Scientific) using the Proxeon nanospray source. Solvent A was water with 0.1% (v/v) formic acid, and solvent B was acetonitrile, 0.1% (v/v) formic acid. The samples were loaded with a constant flow of solvent A at 5 μL/min onto the trapping column. Trapping time was 6 min. Peptides were eluted via the analytical column at a constant flow rate of 0.3 μL/ min, at 40 °C. During the elution step, the percentage of solvent B increased in a linear fashion from 5% to 7% in the first 10 min, then from 7% B to 30% B in the following 105 min, and to 45% B by 130 min. The peptides were introduced into the mass spectrometer via a Pico-Tip Emitter 360 μm OD × 20 μm ID; 10 mm tip (New Objective), and a spray voltage of 2.2 kV was applied. The capillary temperature was set at 300 °C. Full scan MS spectra with a mass range of 375–1500 $m/z$ were acquired in profile mode in the Orbitrap with a resolution of 60000 FWHM using the quad isolation. The RF on the ion funnel was set to 40%. The filling time was set to a maximum of 100 ms with an AGC target of $4 \times 10^5$ ions and 1 microscan. The peptide monoisotopic precursor selection was enabled, along with relaxed restrictions if too few precursors were found. The most intense ions (instrument operated for a 3 s cycle time) from the full scan MS were selected for MS2, using quadrupole isolation and a window of 1 Da. HCD was performed with a collision energy of 35%. A maximum fill time of 50 ms for each precursor ion was set. MS2 data were acquired with a fixed first mass of 120 $m/z$ and acquired in the ion trap in Rapid scan mode. The dynamic exclusion list was set with a maximum retention period of 60 s and a relative mass window of 10 ppm. For the MS3, the precursor selection window was set to the range 400–2000 $m/z$, with an exclusion width of 18 $m/z$ (high) and 5 $m/z$ (low). The most intense fragments from the MS2 experiment were co-isolated (using Synchronus Precursor Selection = 8) and fragmented using HCD (65%). MS3 spectra were acquired in the Orbitrap over the mass range of 100–1000 $m/z$, and the resolution was set to 30,000 FWHM. The maximum injection time (IT) was set to 105 ms, and the instrument was set not to inject ions for all available parallelizable times. The Xcalibur v4.0 and Tune v2.1 were used to acquire and process raw data.

## Parallel reaction monitoring (PRM) for ubiquitin peptides

Ten peptides for quantification of ubiquitin-chain linkages and total ubiquitin (Supplementary Data 6) were selected from ref. [64], and their isotopically labeled version (heavy arginine (U-13C6; U-15N4) or lysine (U-13C6; U-15N2) at the C-term was added), synthesized by JPT Peptide Technologies GmbH (Berlin, Germany) as SpikeTides TQL quality grade. Lyophilized peptides were reconstituted in 20% (v/v) acetonitrile, 0.1% (v/v) formic acid to a final concentration of 0.05 nmol/μL and further pooled together in a ratio 1:1. The peptides mixture was used to make aliquots of 1 pmol/μL in 5% ACN in 50 mM AmBic and stored at −20°. To generate PRM assays, pooled peptides corresponding to ~100 fmol per peptide were digested using LysC and trypsin to cleave the Qtag similar to how described in "Sample preparation for total proteome and analysis of PTMs" with the only difference that 10% formic acid was used instead of 10% TFA for the peptides acidification before the desalting step. Peptides were resuspended at a final concentration of 2.5 fmol/μL and 1 μL was analyzed by both DDA and DIA LC-MS/MS and used for assay generation using Spectrodive v.11.10.2 (Biognosys AG, Schlieren, Switzerland).

For PRM measurements, digested peptides from brain and iNeurons samples were spiked before digestion with AQUA peptides at a concentration of 20 fmol for 1 μg of estimated protein extract, and separated using a nanoAcquity UPLC MClass system (Waters, Milford, MA, USA) with trapping (nanoAcquity Symmetry C18, 5 μm, 180 μm × 20 mm) and an analytical column (nanoE MZ HSS C18 T3 1.8 μm, 75 μm × 250 mm). The outlet of the analytical column was coupled directly to an Orbitrap Fusion Lumos (Thermo Fisher Scientific, Waltham, MA, USA) using the Proxeon nanospray source. Solvent A was water, 0.1% (v/v) formic acid, and solvent B was acetonitrile, 0.1% (v/v) formic acid. Peptides were eluted via the analytical column with a constant flow of 0.3 μL/min. During the elution step, the percentage of solvent B increased nonlinearly from 0% to 25% in 15 min, then from 25% to 50% in 1 min. Total run time was 30 min, including clean-up and column re-equilibration. PRM acquisition was performed in a scheduled fashion for the duration of the entire gradient using the "tMSn" mode with the following settings: resolution 120,000 FWHM, AGC target $3 \times 10^6$, maximum IT 350 ms, isolation window 0.4 $m/z$. For each cycle, a "full MS" scan was acquired with the following settings: resolution 120,000 FWHM, AGC target $3 \times 10^6$, maximum IT 10 ms, scan range 350–1650 $m/z$. Peak group identification was performed using SpectroDive and manually reviewed. Quantification was performed using a spike-in approach using the ratio between endogenous (light) and reference (heavy) peptides for absolute quantification. All the AQUA peptides for total ubiquitin and different types of linkages were quantified except K29.

## Immunoblot and dot-blots

iNeurons or mouse brains were lysed as described in "Sample preparation for total proteome and analysis of PTMs." Protein concentration was estimated by Qubit assay (Invitrogen, Q33211), and 30 μg of protein was used. 4× loading buffer (1.5 M Tris pH 6.8, 20% (w/v) SDS, 85% (v/v) glycerin, 5% (v/v) β-mercaptoethanol) was added to each sample and then incubated at 95 °C for 5 min. For immunoblots, proteins were separated on 4–20% Mini-Protean® TGX™ Gels (BioRad, 4561096) by sodium dodecyl sulfate-polyacrylamide gel electrophoresis (SDS-PAGE) using a Mini-Protean® Tetra Cell system (BioRad, Neuberg, Germany, 1658005EDU). Proteins were transferred to a nitrocellulose membrane (Carl Roth, 200H.1) using a Trans-Blot® Turbo™ Transfer Starter System (BioRad, 1704150). For Dot blot, lysates were directly vacuumed on the nitrocellulose membrane using the Minifold®I Dot-Blot System, 96 Dots (Schleicher & Schuell, SRC-96/1). Membranes were stained with Ponceau S (Sigma, P7170-1L) for 5 min on a shaker (Heidolph Duomax 1030), washed with Milli-Q water, imaged on a Molecular Imager ChemiDocTM XRS + Imaging system (BioRad), and destained by 2 washes with PBS and 2 washes in TBST

(Tris-buffered saline (TBS, 25 mM Tris, 75 mM NaCl), with 0.5% (v/v) Tween 20) for 5 min. After incubation for 5 min in EveryBlot blocking buffer (Biorad, 12010020), membranes were incubated overnight with primary antibodies against total ubiquitin conjugates (Enzo Life Sciences, FK2, BML-PW8810), SQSTM1 (Abcam, ab91526), ubiquitin lys48-Specific (MilliporeSigma, 05-1307), LC3 (Cell Signaling Technology, 2775) or α-tubulin (Sigma, T9026) diluted (1:1000) in enzyme dilution buffer (0.2% (w/v) BSA, 0.1% (v/v) Tween 20 in PBS) at 4 °C on a tube roller (BioCote® Stuart® SRT6). Membranes were washed three times with TBST for 10 min at room temperature and incubated with horseradish peroxidase-coupled secondary antibodies (Dako, P0448/ P0447) at room temperature for 1 h (1:2000 in 0.3% (w/v) BSA in TBST). After three more washes for 10 min in TBST, chemiluminescent signals were detected using ECL (enhanced chemiluminescence) Pierce detection kit (Thermo Fisher Scientific, Waltham, MA, USA, #32109). Signals were acquired on the Molecular Imager ChemiDocTM XRS + Imaging system and analyzed using the Image Lab 6.1 software (Biorad). Membranes were stripped using stripping buffer (1% (w/v) SDS, 0.2 M glycine, pH 2.5), washed 3 times with TBST, blocked, and incubated with the second primary antibody, if necessary.

## Proteasome activity assay

Snap-frozen brain samples were thawed and transferred into 2.0 mL Eppendorf tubes containing PBS. The volume of PBS added was calculated based on an estimated protein content (5% of fresh tissue weight) to reach a 20 µg/µL concentration. Tissues were homogenized using a 5 mL douncer homogenizer, transferred to a new 1.5 mL Eppendorf tube and centrifuged at 17,000 × g for 1 min at 4 °C. Supernatants were then collected and aliquoted to new 1.5 mL Eppendorf tubes and stored at −80 °C. To 25 µL homogenates, 75 µL of 1X lysis buffer was added, which was supplied with the UBPBio's Proteasome Activity Fluorometric Assay Kit II (Cat. # J4120). The homogenates were lysed by sonication in a Bioruptor Plus with high intensity for 60 s ON/30 s OFF at 4 °C for 5 cycles. The lysates were then transferred to a new 1.5 mL Eppendorf tube, and the protein concentration was measured using Nanodrop and Coomassie staining. Lysates corresponding to 50 µg of proteins were used to measure the chymotrypsin-like activity (CT-L) of the proteasome using Suc-LLVY-AMC fluorogenic peptides supplied with the kit as per the manufacturer's protocol. The CT-L activity was determined as the difference between the activity of protein lysates and the residual activity of the lysate in the presence of 300 µM MG132 supplied with the kit. Fluorescence was measured in the kinetic mode at 37 °C for 30 min by TECAN kinetic analysis (excitation 360 nm, emission 460 nm, 1 min read interval) on a Safire II microplate reader (TECAN).

## RNA isolation for RNA-seq analysis

Individual brains from the mice were collected and snap-frozen in liquid nitrogen. On the day of the experiment, samples were thawed on ice, 1 mL of ice-cold Qiazol (Qiagen, 79306) reagent was added and transferred into Precellys® lysing kit tubes (Keramik-kit 1.4/2.8 mm, 2 mL (CKM)) Tissues were homogenized twice at 6000 rpm for 30 s using Precellys® 24 Dual (Bertin Instruments, Montigny-le-Bretonneux, France), transferred to new 2 mL Eppendorf tubes and then further processed using 1 mL syringes with 26 G needles. The RNA extraction was carried out according to the manufacturer's instructions. RNA integrity number was checked at the end of the procedure. Samples were further processed by GENEWIZ from Azenta Life Sciences from the library preparation to the bioinformatic analysis. RNA-seq was performed using Illumina NovaSeq 2 × 150 bp sequencing, 10 M read pairs, and PolyA selection with ERCC spike-in.

## Data processing for mass spectrometry DIA data

Spectral libraries were created for the PTMs by searching DIA and/or DDA runs using Spectronaut Pulsar (18.2.33, Biognosys, Zurich,

Switzerland), while the iNeurons proteome library was generated with Spectronaut Pulsar vr. 16.2.22, as referred to in Supplementary Table 1. The data were searched against species-specific protein databases with an appended list of common contaminants. The data were searched with the following modifications: carbamidomethyl (C) as fixed modification, and oxidation (M), acetyl (protein N-term), lysine di-glycine (K-ε-GG), phosphorylated tyrosine (T) and serine (S), and acetyl-lysine (K-Ac) as variable modifications for the respective PTMs enrichments. A maximum of 3 missed cleavages were allowed for K-Ac and K-ε-GG modifications, and 2 missed cleavages were allowed for phospho enrichment. Killifish MS data from ref. [40] were reanalyzed with the same Spectronaut version using the same settings. DIA PTMs data were searched against their respective spectral library using Spectronaut Professional (18.2.33, Biognosys, Zurich, Switzerland). The library search was set to a 1 % false discovery rate (FDR) at both protein and peptide levels. Data were filtered for Qvalue percentile of 0.2, and global imputation and true automatic normalization were applied. For the DIA proteome iNeurons data, no imputation was used, and local normalization was applied. Relative quantification was performed in Spectronaut for each pairwise comparison using the replicate samples from each condition using default settings. Candidates and report tables were exported from Spectronaut for downstream analysis. The liver and dietary intervention datasets were processed using direct-DIA search (library-free) using the parameters described in Supplementary Table 1. Tests for differential abundance were performed using an unpaired t-test between replicates. P values were corrected for multiple testing using the method described by Storey[107] to obtain FDR-adjusted P values (Q values). Candidates and report tables were exported from Spectronaut and used for downstream analysis.

## Data processing for TMT-labeled samples

For the mouse brain proteome, TMT-10plex data were processed using ProteomeDiscoverer v2.0 (Thermo Fisher). Data were searched against the relevant species-specific fasta database (Uniprot database, Swissprot entry only, release 2016_01 for *Mus musculus*) using Mascot v2.5.1 (Matrix Science) with the following settings: enzyme was set to trypsin, with up to 1 missed cleavage. MS1 mass tolerance was set to 10 ppm and MS2 to 0.5 Da. Carbamidomethyl cysteine was set as a fixed modification, and oxidation of Methionine as a variable. Other modifications included the TMT-10plex modification from the quantification method used. The quantification method was set for reporter ions quantification with HCD and MS3 (mass tolerance, 10 ppm). The FDR for peptide-spectrum matches (PSMs) was set to 0.01 using Percolator[108]. Reporter ion intensity values for the PSMs were exported and processed with procedures written in R (v. 3.4.1), as described in ref. [109]. Briefly, PSMs mapping to reverse or contaminant hits, or having a Mascot score below 15, or having reporter ion intensities below $1 \times 10^3$ in all the relevant TMT channels were discarded. TMT channel intensities from the retained PSMs were then log2 transformed, normalized, and summarized into protein group quantities by taking the median value. At least two unique peptides per protein were required for the identification, and only those peptides with one missing value across all 10 channels were considered for quantification. Protein differential expression was evaluated using the limma package[110]. Differences in protein abundances were statistically determined using an unpaired Student's t-test moderated by the empirical Bayes method. P values were adjusted for multiple testing using the Benjamini−Hochberg method (FDR, denoted as "Adj.*P*")[111]. Proteins with adj p < 0.05 were considered as significantly affected.

## Modified peptide abundance correction

For each enrichment, PTM report tables were exported from Spectronaut. Correction factors were calculated using the aging proteome data to correct the quantities of modified peptides for underlying changes in protein abundance across the age groups compared. For

each condition and protein group, the median protein quantity was calculated and then divided by the median protein quantity in the young age group. Each modified peptide was matched to the correction factor table by the protein identifier. If a modified peptide was mapped to 2 or more proteins, the correction factor was calculated using the sum of the quantities of these proteins. Further, the correction was carried out by dividing peptide quantities by the mapped correction factors and log2 transforming. Differences in peptide quantities were statistically determined using the $t$-test moderated by the empirical Bayes method as implemented in the R package limma[110]. The number and percentage of corrected sites for each dataset are reported in Supplementary Table 2.

### Data processing for RNA-Seq

Data processing was carried out by Azenta Life Sciences. In summary, Trimmomatic v.0.36 was used to trim sequence reads by removing potential adapter sequences and low-quality bases. The cleaned reads were then aligned to the Mus musculus GRCm38 reference genome from ENSEMBL using the STAR aligner v.2.5.2b. STAR, being a splice-aware aligner, identifies splice junctions and incorporates them to accurately map full-length reads. This alignment process generated BAM files. To quantify unique gene hit counts, featureCounts from the Subread package v.1.5.2 was employed. Counts were obtained based on the gene_id attribute in the annotation file, considering only uniquely mapped reads located in exon regions. For strand-specific libraries, read counting was performed accordingly. The resulting gene count matrix was used for downstream analysis of differential gene expression. DESeq2 was used to compare gene expression across user-defined sample groups. The Wald test was applied to calculate $p$ values and log2 fold changes.

### GO enrichment analysis

Gene Set Enrichment Analysis (GSEA) was performed using the R package clusterProfiler[112], using the function gseGO. Briefly, protein entries were mapped to the human gene name orthologues and given as input to the function to perform the enrichment. For the GO term ORA, the topGO R package was used.

### Alignment of ubiquitylated sites

To align ubiquitylated sites between killifish and mouse, proteins with at least one ubiquitylated site in killifish were chosen. A local alignment was conducted using protein BLAST (v2.12.0+)[113] with default parameters between the killifish protein sequences (Nfu_20150522, annotation nfurzeri_genebuild_v1.150922) against the mouse Uniprot proteome (release 2021_04). The top 10 matches from the BLAST search were retrieved, and each modified lysine was placed into the local alignment to determine the corresponding position in the mouse protein. A specific lysine was deemed conserved if there was a lysine at the corresponding mouse position in at least one of the top 10 hits from the BLAST alignment. A ubiquitylation site was considered conserved if the same lysine mapped from killifish to mouse was also identified as ubiquitylated in the mouse dataset. The same approach was used to map mouse and killifish ubiquitylated residues onto the human proteome (release 2021_04).

### Reporting summary

Further information on research design is available in the Nature Portfolio Reporting Summary linked to this article.

## Data availability

The proteomics and sequencing data have been deposited with the following identifiers: Ubiquitylation of mouse brain aging: MSV000093686, Acetylation of mouse brain aging: MSV000093689, Phosphorylation of mouse brain aging: MSV000093687, Whole proteome mouse brain aging: MSV000093690, RNA-seq mouse brain aging: GSE253375, AQUA-PRM ub-chains brain aging mouse: MSV000093996, Ubiquitylation of mouse liver aging: MSV000096232, Whole proteome mouse liver aging: MSV000096231, Ubiquitylation in iNeurons: MSV000093691, Whole proteome iNeurons: MSV000093693, AQUA-PRM ub-chains iNeurons: MSV000096233, Ubiquitylation mouse brain dietary intervention: MSV000096229, Whole proteome mouse brain dietary intervention: MSV000096226. Source data are provided with this paper.

## Code availability

The code and R package are available on Zenodo (10.5281/zenodo.15270436).

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

## Acknowledgements

The authors gratefully acknowledge support from the FLI Core Facilities, Proteomics and the Fish and Mouse Facilities. A.O. is supported by the German Research Council (Deutsche Forschungsgemeinschaft, DFG) via the Research Training Group ProMoAge (GRK 2155), the Else Kröner Fresenius Stiftung (award number: 2019_A79), the Fritz-Thyssen Foundation (award number: 10.20.1.022MN), the Chan Zuckerberg Initiative Neurodegeneration Challenge Network (award numbers: 2020-221617, 2021-230967, and 2022-250618), and the NCL Stiftung. The FLI is a member of the Leibniz Association and is financially supported by the Federal Government of Germany and the State of Thuringia.

## Author contributions

Conceptualization: A.M., D.D.F., A.O. Data curation: A.M., D.D.F. Investigation: A.M., D.P., A.K.S., A.l.M., O.O. Methodology: A.M., E.C. Project administration: A.O. Data analysis: A.M., D.D.F. Supervision: A.O. Visualization: A.M., D.D.F. Writing—original draft: A.M., D.D.F., A.O. Writing—review & editing: D.P.

## Funding

## Competing interests

The authors declare no competing interests.
