## [Transparent Peer Review file · Nature Communications]

Aging and diet alter the protein ubiquitylation landscape in the mouse brain

Corresponding Author: Dr Alessandro Ori

Version 0:

Reviewer comments:

Reviewer #1

(Remarks to the Author)

The authors have addressed all my comments and I recommend publication of the manuscript

Reviewer #2

(Remarks to the Author)

The authors did a great job with addressing comments. The revised manuscript is now much more clear and scientifically precise. I am looking forward to see the manuscript published. It brings across an important message about ubiquitylation and aging.

Minor comments that the authors may want to consider:

Introduction is too long (different proteasostasis pathways do not need to be explained)

I think that the unprecise statement as "autophagy declines with aging" is below the level of this work. There are many forms of selective autophagy that have not all been investigated in all tissues. Then at least the authors should on page 2 mention "brain" in the sentence. There are indeed studies showing that autophagy can be induced as a compensatory effect to declining UPS (that will also depend on the aging stage). I think that the aging field would benefit from avoiding this broad conclusion.

I would remove additional explanations why p62 can be used both as autophagy and proteasome markers as this is confusing for the reader. I would actually remove p62 as a control for proteasome inhibitor and only leave the accumulation of Ub conjugates.

Mention somewhere the connection between USP33 and PARKIN deubiquitylation.

Add in Figure 1 also a volcano plot for aging-induced proteome changes showing that Ub is much more affected.

Reviewer #3

(Remarks to the Author)

The study by Marino and colleagues undertakes an analysis of changes in posttranslational modification over the course of age. Their study runs the gambit of mice, to killifish to iPSCs. They make a number of highly interesting findings in their work that will be a useful resource to other researchers.

I was a reviewer of this article when it was under consideration at a prior journal before it was transferred to Nature Communications. In my original review I was highly critical. I provided a number of strong criticisms. I am happy to say that the authors rose to the challenge. They appear to have put in a huge amount of work to address all of the issues I raised in my original review. I think they have done a tremendous amount of work making the manuscript substantially stronger and are to be commended! I think this is a strong, insightful, and impactful study.

Marino. et al. Response to Review round: 2

Reviewer #1:

The authors have addressed all my comments and I recommend publication of the manuscript

We thank the reviewer for positively assessing our work and review process!

Reviewer #2:

The authors did a great job with addressing comments. The revised manuscript is now much more clear and scientifically precise. I am looking forward to see the manuscript published. It brings across an important message about ubiquitylation and aging.

Minor comments that the authors may want to consider:

Introduction is too long (different proteasostasis pathways do not need to explained)

I think that the unprecise statement as "autophagy declines with aging" is below the level of this work. There are many forms of selective autophagy that have not all been investigated in all tissues. Then at least the authors should on page 2 mention "brain" in the sentence. There are indeed studies showing that autophagy can be induced as compensatory effect to declining UPS (that will also depend on the aging stage). I think that the aging field would benefit from avoiding this broad conclusions.

I would remove additional explanations why p62 can be used both as autophagy and proteasome markers as this is confusing for the reader. I would actually remove p62 as control for proteasome inhibitor and only leave the accumulation of Ub conjugates.

Mention somewhere connection between USP33 and PARKIN deubiquitylation.

Add in Figure 1 also volcano plot for aging induced proteome changes showing that Ub is much more affected.

We thank the reviewer for their suggestions that helped us improve our manuscript. According to the reviewer's comment, we modified the introduction, making it shorter and removing statements on autophagy that were too broad.

Manuscript, page 2: "Post-translational modifications (PTMs) expand the chemical diversity of proteins by generating distinct proteoforms encoded by the same gene ¹. PTMs modulate proteins' localization, interactions, stability, and turnover, thereby influencing protein homeostasis (proteostasis) ². Loss of proteostasis is a hallmark of aging ³⁻⁵ and age-related diseases, particularly neurodegenerative disorders ^{6,7}. Acetylation, phosphorylation, and ubiquitylation are among the most studied PTMs, accounting for more than 90% of all reported modifications to date ⁸. Acetylation is fundamental for brain gene expression, regulating the accessibility of histones ⁹. Histone acetylation is essential for neuronal maturation, synapse formation, and establishment of neuronal circuits ¹⁰. Phosphorylation mediates signaling and modulates multiple neuronal functions. For instance, calcium/calmodulin-dependent protein kinase II (CaMKII) modulates synaptic strength, mainly by affecting

trafficking, function, and anchoring to the postsynaptic membrane of glutamate transmembrane receptors ¹¹. Ubiquitylation plays a central role in protein degradation through the ubiquitin-proteasome system (UPS) ¹². Decline of UPS activity is an early event during brain aging ¹³. Ubiquitin signaling is also required for the turnover of organelles, e.g., mitochondria via mitophagy and endoplasmic reticulum (ER) via ER-phagy ^{14,15}, and it can modulate synaptic activity and plasticity ^{16,17}.

Proteins carrying PTMs, including acetylation, phosphorylation, and ubiquitylation, have been found in protein aggregates in samples from patients suffering from different types of neurodegenerative diseases ^{18–20}. For instance, hyperphosphorylation and ubiquitylation of the microtubule-associated protein Tau (MAPT) are characteristic of Alzheimer's disease ^{18,21}. Similarly, the RNA-binding protein TDP-43 is ubiquitylated and hyperphosphorylated when mislocalized to the cytoplasm in frontotemporal dementia (FTD) ^{22,23}, while GFAP acetylation has been associated with amyotrophic lateral sclerosis (ALS) ²⁴.

A few previous studies investigated proteome-wide changes in PTMs during vertebrate brain aging. Age- and tissue-specific changes in cysteine oxidation have been described ²⁵, e.g., in tRNA aminoacylation complexes, and found to do not correlate with protein abundance changes. Protein persulfidation, another redox modification, was shown to decrease during rat brain aging and in neurodegenerative disorders ²⁶. Additionally, in rats, phosphorylation data showed that specific phosphosite levels changed in the aging brain due to mislocalized protein kinases ²⁷. These studies proved that altering PTMs might contribute to the loss of protein homeostasis in aging and age-related neurodegenerative disorders. However, a systematic investigation of major PTMs during physiological brain aging is still lacking.

To fill this knowledge gap, we quantified the effect of aging on protein acetylation, phosphorylation, and ubiquitylation in the mouse brain using mass spectrometry (MS). We found ubiquitylation to be the most affected PTM. Given the conservation of ubiquitin across evolution ²⁸, we asked whether similar alterations could be observed in different species using the short-lived killifish *Nothobranchius furzeri*. We chose killifish because of its spontaneous age-related brain phenotypes that are common to human neurodegeneration, including accumulation of phosphorylated MAPT with aging ^{29–31}. By combining mouse and killifish data, we were able to define an ubiquitylation aging signature conserved in the brains of these two species. To identify the causes of altered protein ubiquitylation in the aged brain, we used human induced pluripotent stem cell (iPSC)-derived neurons (iNeurons) and showed that more than one-third of the observed age-related changes in ubiquitylation can be attributed to a decline in proteasome activity. Finally, we tested whether a

dietary intervention applied to old mice can influence protein ubiquitylation in the brain and reverse some of the effects of aging. ”

As suggested to avoid confusion, we removed the p62 explanation from the text and the immuno blot from Figure S6C.

Revise Figure S6C. Immunoblots for K48 ubiquitin chains and LC3-II/LC3-I. Representative blots from N=3 biological replicates.

We now mention the connection between USP33 and Parkin deubiquitylation in the paragraph:

Manuscript, page 14: “Another notable instance of decreased ubiquitylation in our dataset involves USP33, a deubiquitylase localized to the mitochondrial outer membrane and known to interact with Parkin. USP33 depletion has been shown to increase K63-linked ubiquitin conjugates on Parkin, leading to its stabilization and enhanced mitophagy 83 Thus, an age-related decline in ubiquitylation could stabilize and activate USP33, potentially modulating Parkin-dependent mitophagy. Altered mitophagy might, in turn, contribute to the elevated ubiquitylation of mitochondrial proteins observed in aged brains (Figure 1E). ”

Reviewer #3:

The study by Marino and colleagues undertakes an analysis of changes in posttranslational modification over the course of age. Their study runs the gambit of mice, to killifish to iPSCs. They make a number of highly interesting findings in their work that will be a useful resource to other researchers.

I was a reviewer of this article when it was under consideration at a prior journal before it was transferred to Nature Communications. In my original review I was highly critical. I provided a number of strong criticisms. I am happy to say that the authors rose to the challenge. They

appear to have put in a huge amount of work to address all of the issues I raised in my original review. I think they have done a tremendous amount of work making the manuscript substantially stronger and are to be commended! I think this is a strong insightful and impactful study.

We thank the reviewer for the praiseful words and we are pleased they find it a well-executed and impactful study.